# PROVABLE LENGTH GENERALIZATION IN SEQUENCE PREDICTION VIA SPECTRAL FILTERING

## ABSTRACT

We consider the problem of length generalization in sequence prediction. We define a new metric of performance in this setting – the Asymmetric-Regret– which measures regret against a benchmark predictor with longer context length than available to the learner. We continue by studying this concept through the lens of the spectral filtering algorithm. We present a gradient-based learning algorithm that provably achieves length generalization for linear dynamical systems. We conclude with proof-of-concept experiments which are consistent with our theory.

## 1 INTRODUCTION

Sequence prediction is fundamental to machine learning, with applications in NLP, forecasting, and control systems. In this setting, a learner observes a sequence of tokens and iteratively predicts the next token, suffering a loss that measures the discrepancy between the predicted and the true token. Predicting sequences is vital for tasks like language modeling and autonomous control.

A key challenge in sequence prediction is understanding the role of *context length*—the number of previous tokens used to make the upcoming prediction—and designing predictors that perform well with limited context due to computational and memory constraints. These resource constraints become particularly significant during the training phase of a predictor, where the computational cost of using long sequences can be prohibitive. Thus, designing predictors that learn from shorter contexts but generalize to longer ones is crucial. This leads us to the central question of our investigation: Can we develop algorithms that learn effectively using short contexts but perform comparably to models that use longer contexts?

To address this question, we introduce a new performance metric—Asymmetric-Regret—which measures the difference in total prediction loss between an online predictor with limited context length and a benchmark predictor with a longer context. Unlike classical regret, which assumes both the learner and the benchmark operate under the same conditions, Asymmetric-Regret accounts for the asymmetry in context lengths, providing a more realistic assessment of performance in resource-constrained settings. With a formal and well-defined notion of Asymmetric-Regret in hand, we begin our investigation with the following question: are there algorithms that can attain non-trivial bounds on the Asymmetric-Regret for natural sequences?

We explore this concept through the lens of spectral filtering algorithms (Hazan et al., 2017b; 2018). Spectral filtering has emerged as a robust method for learning linear dynamical systems when the system is unknown and the hidden state is unobserved. Linear dynamical systems are a useful and rich class to study. Although they are applicable in many domains, they have been particularly useful in large language modeling applications. Since next-token generation is a sequence prediction problem, these methods are naturally applicable as a building block to use as layers in LLMs. Methods which are designed to solve sequence prediction in linear dynamical systems have been used to design state space models which have achieved SOTA performance on many LLM tasks, with efficiency gains during both training and inference Gu et al. (2021b); Poli et al. (2023); Gu & Dao (2023). Spectral filtering was introduced in Hazan et al. (2017a) as a method which provably achieves $O(\sqrt{T})$ regret when compared with the best LDS predictor (without any assumptions on the sequence data). Beyond their theoretically sound properties, spectral filtering-based predictors have proven practical in recent applications. Notably, the Spectral Transform Unit (Agarwal et al., 2023), a neural architecture built using spectral filtering, has recently shown promise on sequence prediction over a range of modalities (Liu et al., 2024).

In this work, we extend the theoretical understanding of spectral filtering by demonstrating that these predictors can achieve length generalization. Specifically, we present a gradient-based online learning algorithm for spectral filtering and show that we can learn and make predictions on a smaller context length while still achieving the same regret bounds as if we had used a much longer context length. Formally, we prove that this algorithm guarantees Asymmetric-Regret $\tilde{O}(\sqrt{T})$.

Beyond theoretical interest, our work is practically motivated by challenges in length generalization faced by large language models (LLMs). As previously mentioned, methods which emerged from studying linear dynamical systems have proven useful in LLMs, including spectral filtering and the Spectral Transform Unit. LLMs struggle to generalize to longer sequences than those seen during training (Abbe et al., 2023; Anil et al., 2022; Jelassi et al., 2023; Zhou et al., 2023; Delétang et al., 2022; Dziri et al., 2024; Zhou et al., 2024) and extensive empirical research has been dedicated to addressing this (Kazemnejad et al., 2024; Shen et al., 2023; Dai, 2019; Chi et al., 2022; Li et al., 2023; Press et al., 2021). Despite its importance and extensive empirical research, provable theoretical results on length generalization remain largely elusive. We view our work as a step toward addressing this gap. The asymmetric regret bounds we establish in this paper imply that spectral filtering is able to implicitly handle the difficult problem of deciding how to use and store tokens much earlier in a sequence for next-token prediction. Since most empirical methods introduced to improve length generalization are task-specific, this is an exciting feature. It suggests that simply incorporating spectral filtering into neural architectures may have the potential to improve length generalization.

## 1.1 OUR CONTRIBUTIONS

Consider **online sequence prediction** in which the predictor iteratively receives input $u_t \in \mathcal{R}^{d_{in}}$ and then makes a prediction $\hat{y}_t \in \mathcal{R}^{d_{out}}$ of the output, after which the true output $y_t$ is revealed. The goal of the predictor is to minimize error according to a given convex and Lipschitz loss function $\ell_t(y_t, \hat{y}_t)$. In this work we consider the class of *spectral filtering* predictors, introduced by Hazan et al. (2017b). A spectral filtering predictor is characterized by parameters $(T, M_{i\,i=1}^k, k)$ and outputs predictions $\hat{y}_t$ of the form

$$\hat{y}_t = y_{t-1} + \sum\nolimits_{i=1}^k M_i u_{(t-1):0} \phi_i,$$

where $u_{(t-1):0} \in \mathbb{R}^{d_{in} \times T}$ is a matrix whose columns are the previous inputs $u_{t-1}, u_{t-2}, \ldots, u_0$ (possibly zero-padded as necessary), $\{\phi_j\}_{j=1}^k$ are the $T$-dimensional spectral filters, $\{M_i\}_{i=1}^k \subset \mathcal{R}^{d_{out} \times d_{in}}$ are matrices which are learned online, and $k$ is the number of filters used. Hazan et al. (2017b) provide an algorithm to learn $\{M_i\}_{i=1}^k$ and show this achieves nearly optimal regret bounds when measured against the best Linear Dynamical System (LDS) predictor. We explore whether the full history $u_{(t-1):0}$ is needed to learn $\{M_i\}_{i=1}^k$. More broadly, we explore whether predictor classes and corresponding online learning algorithms exist that can achieve context length generalization—that is, they use only a short recent history during learning but perform nearly as well as if they had used the full, much longer history length. Of course, predictors which perform poorly on systems that require long memory can trivially achieve context length generalization if their performance is poor regardless of the context length used. Notably, spectral filtering predictors excel in systems with long memory (Hazan et al., 2017b).

To properly understand context length generalization, we introduce the notion of *Asymmetric-Regret*. The idea is to consider the regret of learning a predictor from a class which is only allowed to use context length $L'$ against the best predictor which is allowed to use (potentially much longer and therefore asymmetric) context length $L$. Let $\Pi_L$ denote the class of predictors in $\Pi$ which use context length $L$. Given an algorithm $\mathcal{A}(L')$ which learns over predictors from some class $\Pi_{L'}$, the Asymmetric-Regret over horizon $T$ is

$$\text{Regret}_{\text{Asymmetric},T}\left(\mathcal{A}(L'), \Pi_L\right) \stackrel{\text{def}}{=} \sum\nolimits_{t=1}^T \ell_t(y_t, \hat{y}_t^{\mathcal{A}(L')}) - \min_{\pi \in \Pi_L} \ell_t(y_t, \hat{y}_t^\pi).$$

Our first result shows that spectral filtering generalizes from a history of $T^q$, where $q \in [0, 1]$, to $T$ for certain linear dynamical systems. It is formally given in the following theorem.

**Theorem 1.** *Let $T \in \mathbb{Z}_{>0}$ and $q \in [0, 1]$. Consider a sequence $(y_1, \ldots, y_T)$ generated by an unknown and noiseless linear dynamical system defined by matrices $(A, B, C, D)$ as per Eq. 1,*

*Assume the input sequence $u_{0:(t-1)}$ is sufficiently well-conditioned, satisfying $\sum_{t=0}^{T-1}(T-t)u_t u_t^\top \succeq \left(\frac{2|C||B|}{\sqrt{T}}\right) I$. Suppose the eigenvalues of $A$ lie within the range $\left[0, 1 - \frac{\log(T)}{8T^q}\right] \cup \left[1 - \frac{1}{2T^{5/4}}, 1\right]$.*

*Let $\mathcal{A}(L)$ denote Algorithm 1 operating with context length $L$, and let $\Pi_L^{\text{SF}}$ denote the class of spectral filtering predictors using context length $L$. For the squared loss $\ell_t(y, y') = |y - y'|^2$ and sufficiently large $T$, it holds that:*

$$Regret_{\text{Asymmetric},T}\left(\mathcal{A}(T^q), \Pi_T^{\text{SF}}\right) \leq \tilde{O}(\sqrt{T}).$$

This theorem indicates that for any $q \in [0, 1]$, the Asymmetric-Regret is bounded by $\tilde{O}(\sqrt{T})$. However, as $q$ decreases, the class of linear dynamical systems for which this bound holds becomes more restricted due to the eigenvalue conditions on $A$. The spectrum of $A$ determines the memory of the system; when the eigenvalues of $A$ are 1, the system is only marginally-stable and standard predictors which aim to use low memory typically fail. Critically, Theorem 1 holds even for these marginally-stable systems. When interpreting this result, it's important to note that the class of spectral filtering predictors $\Pi_T^{\text{SF}}$ which use the full context length are provably able to predict well on marginally-stable Linear Dynamical Systems (Hazan et al., 2017b)[1]. Therefore, this result implies that spectral filtering predictors are able to context length generalize in a nontrivial way.

Inspired by the way in which Theorem 1 is sensitive to the spectrum of $A$, we develop a novel variation on the Spectral Filtering algorithm, presented in Algorithm 2, which achieves robust length generalization without added assumptions on the spectrum of $A$ (whenever the context-length is at least $T^{1/4}$). Algorithm 2 achieves this by using two autoregressive components $y_{t-1}$ and $y_{t-2}$ to construct its prediction $\hat{y}_t$ of $y_t$. We provide our main theorem of this work.

**Theorem 2.** *Let $T \in \mathbb{Z}_{\geq 0}$ and $q \in \left[\frac{1}{4} + \frac{\log(\log(T)/8)}{\log(T)}, 1\right]$. Consider a sequence $(y_1, \ldots, y_T)$ generated by an unknown and noiseless linear dynamical system defined by matrices $(A, B, C, D)$ as per Eq. 1. Assume the input sequence $u_{0:(t-1)}$ is sufficiently well-conditioned, satisfying $\sum_{t=0}^{T-1}(T-t)u_t u_t^\top \succeq \left(\frac{2|C||B|}{\sqrt{T}}\right) I$. Let $\mathcal{A}(L)$ denote Algorithm 2 operating with context length $L$, and let $\Pi_L^{\text{SF}}$ denote the class of spectral filtering predictors using context length $L$. For the squared loss $\ell_t(y, y') = |y - y'|^2$ and sufficiently large $T$, it holds that:*

$$Regret_{\text{Asymmetric},T}\left(\mathcal{A}(T^q), \Pi_T^{\text{SF}}\right) \leq \tilde{O}(\sqrt{T}).$$

Finally, we experimentally confirm the results of Theorem 1 and Theorem 2 on synthetic data generated by an LDS. Interestingly, we find that Theorem 1 accurately predicts when length generalization is possible; indeed, when the data is generated by an LDS which has eigenvalues in the "bad" range $[1 - \log(T)/(8T^q), 1 - 1/(2T^{5/4})]$ we find that the limited context length spectral filtering predictors are unable to length generalize. However, when the data is generated by and LDS which has eigenvalues "hugging" this bad range (i.e. either just smaller than $1 - \log(T)/(8T^q)$ or just larger than $1 - 1/(2T^{5/4})$), the limited context length spectral filtering predictors successfully length generalize, demonstrating the sharpness of our analysis. Next, we see that adding the second autoregressive term allows for robust length generalization on marginally-stable systems with no spectral assumption. Lastly, we conduct experiments using the STU neural architecture to test the hypothesis that this architecture should simply length generalize without any task-specific engineering. We consider the induction heads synthetic task and find that the out-of-the-box STU neural architecture does indeed enjoy some level of length generalization. This suggests that incorporating spectral filtering into neural architectures, like the STU, may provide improved length generalization in deep learning applications. We leave further empirical study on this for future work.

## 1.2 RELATED WORK

The literature for sequence prediction is too broad to survey in detail, so we give a few highlights of the recent rapid advancements. The most notable progress includes the Transformer model (Vaswani

---

[1]The only LDS's for which there can be any useful results are those with $A$'s eigenvalues in $[-1, 1]$, i.e. marginally-stable systems. We recall that the spectral filtering principle can be readily applied to handle negative eigenvalues in $[-1, 0]$ (see Appendix D of Agarwal et al. (2023), for example). For ease of presentation, we focus on capturing the length generalization effects of eigenvalues in $[0, 1]$ in the sequel, and so we suppose without loss of generality that $A \succeq 0$.

et al., 2017) that incorporates an attention mechanism for accurate sequence prediction in many domains (Brown et al., 2020; Dosovitskiy et al., 2020; Jumper et al., 2021). Transformer models and their attention layers have memory/computation requirements that scale quadratically with context length. Many approximations have been proposed (see Tay et al. (2022) for a recent survey).

Motivated by the high memory and compute requirements of transformers, state space models were revisited starting from (Gu et al., 2020; 2021b) who propose and develop the HiPPO theory. Gu et al. (2021a) develop the S4 parameterization to address the bottlenecks of training efficiency, performance and numerical stability. Further works in the area show SOTA performance and include Gupta et al. (2022); Smith et al. (2023); Orvieto et al. (2023); Gu & Dao (2023). State space models are very efficient for training and inference, but can suffer in long-context applications. This motivated the use of spectral filtering technique for learning marginally-stable linear dynamical systems (Hazan et al., 2017b; 2018). This technique was incorporated to a neural architecture in Agarwal et al. (2023), that was recently shown to perform well across several modalities (Liu et al., 2024).

From an applied perspective, generalization in sequence prediction has been recently studied in Hou et al. (2024) through the theoretical lens of Turing programs. They propose a methodology that empirically improves length generalization across a diverse set of tasks. There are also architecture-specific approaches to length generalization such as ALiBi positional embeddings for transformers (Press et al., 2022), but such methods lack provable guarantees and can have varying empirical performance (Kazemnejad et al., 2024).

In contrast, our investigation starts from the theory of regret minimization in games and online learning. Regret minimization has the advantage that it implies generalization in the statistical learning setting (see e.g. Cesa-Bianchi et al. (2004)) and is usually accompanied by efficient algorithms such as online gradient descent (see e.g. Hazan et al. (2016)). Our new notion of Asymmetric-Regret incorporates asymmetric information access between the online learner and the benchmark class.

## 2 BACKGROUND AND SETTING

In the **online sequence prediction** setting the predictor iteratively receives input $u_t$ and makes prediction $\hat{y}_t$ of the output, after which the true output $y_t$ is revealed. The goal is to minimize error according to a given (convex Lipschitz) loss function $\ell_t(y_t, \hat{y}_t)$.

In online learning, we usually do not make statistical assumptions about the generation of the input sequence. As such, performance is measured relative to a certain benchmark class of predictors. A prediction algorithm $\mathcal{A}$ is measured by regret, or difference in total loss, vs. a class of reference predictors $\Pi^{\text{ref}}$ (such as linear predictors), i.e.

$$\text{Regret}_T(\mathcal{A}, \Pi) = \sum_{t=1}^{T} \ell_t(y_t, \hat{y}_t^{\mathcal{A}}) - \min_{\pi \in \Pi} \sum_{t=1}^{T} \ell_t(y_t, \hat{y}_t^{\pi}).$$

This formulation is valid for online sequence prediction of any signal. We are particularly interested in signals that are generated by dynamical systems. A time-invariant linear dynamical system is given by the dynamics equations

$$x_{t+1} = Ax_t + Bu_t + w_t \;\; , \;\; y_{t+1} = Cx_t + Du_t + \zeta_t, \tag{1}$$

where $x_t$ is the (hidden) state, $u_t$ is the input or control to the system, and $y_t$ is the observation. The terms $w_t, \zeta_t$ are noise terms, and the matrices $A, B, C, D$ are called the system matrices.

Many methods exist for linear dynamical systems and their performance guarantees rely heavily on the spectrum of $A$. The system is unstable whenever $|\lambda_{\max}(A)| > 1$ because the norm of the observations tends towards infinity, stable when $|\lambda_{\max}(A)| < 1$, and marginally-stable if $|\lambda_{\max}(A)| = 1$. When $|\lambda_{\max}(A)| = 1 - \delta < 1$, typical methods (i.e. Kalman filtering) must use a history of at least $\gg \frac{1}{\delta}$ previous states to accurately capture the dynamics. As $\delta$ gets smaller (i.e. long memory) it therefore becomes difficult for methods to directly learn these relationships. Methods which learn the system matrices require knowledge of the dimension of the hidden state (which may be very large) and can also be unstable for systems with long memory. Through a particular parameterization and convex relaxation, however, the spectral filtering algorithm is able to efficiently predict observations from marginally-stable systems with sublinear regret. We provide more background on spectral filtering in Section 2.2, and more details on the rich theory of linear dynamical systems may be found in Hazan et al. (2020).

### 2.1 CONTEXT LENGTH GENERALIZATION AND THE ASYMMETRIC-REGRET METRIC

We say that an online predictor has context length $L$ if it bases its prediction $\hat{y}_t$ only on information from the previous $L$ timesteps, i.e. $u_{t:t-L}$ and $y_{t:t-L}$. The key question in our work is whether there are algorithms which learn and predict using a short context length, but perform as well as had they been allowed to use long context length. To formalize this notion, we introduce Asymmetric-Regret whose definition we restate here:

**Definition 3** (Asymmetric-Regret). Let $\Pi_{L'}^{\text{learn}}$ be a class of predictors which use context length $L'$ and let $\Pi_L^{\text{ref}}$ be a reference class of predictors which use context length $L$. The *Asymmetric-Regret* with respect to (convex Lipschitz) loss $\ell_t$ over horizon $T$ of an algorithm $\mathcal{A}(L')$ which tries to learn a predictor from $\Pi_{L'}^{\text{learn}}$ is

$$\text{Regret}_{\text{Asymmetric},T}\left(\mathcal{A}(L'),\Pi_L^{\text{ref}}\right) \stackrel{\text{def}}{=} \sum_{t=1}^{T} \ell_t(y_t, \hat{y}_t^{\mathcal{A}(L')}) - \min_{\pi \in \Pi_L} \sum_{t=1}^{T} \ell_t(y_t, \hat{y}_t^{\pi}).$$

To gain a better understanding of Asymmetric-Regret, note that the typical notion of regret in sequence prediction sets $L' = T$ for the given class of predictors and sets $L = T$ for the given reference class of predictors $\Pi^{\text{ref}}$ by default. In this case Asymmetric-Regret recovers typical regret,

$$\text{Regret}\left(\mathcal{A},\Pi^{\text{ref}}\right) = \text{Regret}_{\text{Asymmetric},T}\left(\mathcal{A}(T),\Pi_T^{\text{ref}}\right).$$

However, if $L' < T$, any upper bound on $\text{Regret}_{\text{Asymmetric},T}\left(\mathcal{A}(L'),\Pi_T^{\text{ref}}\right)$ immediately implies an upper bound on $\text{Regret}\left(\mathcal{A},\Pi^{\text{ref}}\right)$ since the algorithm $\mathcal{A}(T)$ can choose to only use context length $L'$ and ignore the rest. Therefore, Asymmetric-Regret is a stronger notion than typically used.

### 2.2 SPECTRAL FILTERING

Spectral filtering is a notable deviation from the standard theory of linear dynamical systems that allows efficient learning in the presence of arbitrarily long memory (Hazan et al., 2017b). The idea is to project the sequence of inputs to a small subspace that is constructed using the special structure of discrete linear dynamical systems. The output of the spectral filtering predictor is represented as

$$\hat{y}_t = y_{t-1} + \sum_{i=1}^{k} M_i u_{(t-1):0} \phi_i, \tag{2}$$

where $u_{(t-1):0} \in \mathbb{R}^{d_{\text{in}} \times T}$ is a matrix whose columns are the previous inputs $u_{t-1}, \ldots, u_0$ (possibly zero-padded as necessary), $\{\phi_j\}_{j=1}^{k}$ are the $T$-dimensional spectral filters that can be computed offline given the target sequence length $T$, and $\{M_i\}_{i=1}^{k} \subset \mathcal{R}^{d_{\text{out}} \times d_{\text{in}}}$ are the matrices parameterizing the model. These spectral filters are the eigenvectors of the matrix constructed as the average of outer products of the discrete impulse-response functions as we now detail.

Let $\mu_{\alpha,T} = (1-\alpha)[1,\alpha,\alpha^2,...,\alpha^T]$ be the (weighted) impulse-response vector corresponding to a one dimensional linear dynamical system with parameter $\alpha$ unfolded to $T$ time steps, and consider the symmetric matrix

$$H_T \stackrel{\text{def}}{=} \int_0^1 \mu_{\alpha,T} \mu_{\alpha,T}^{\top} d\alpha. \tag{3}$$

Since $H_T$ is a real PSD matrix, it admits a real spectral decomposition, and the (non-negative) eigenvalues can be ordered naturally by their value. Let $\{(\sigma_j \in \mathbb{R}, \phi_j \in \mathbb{R}^L)\}_{j=1}^{L}$ be the eigenvalue-eigenvector pairs of $H_T$ ordered to satisfy $\sigma_1 \geq \sigma_2 \geq \ldots \geq \sigma_d$. The spectral filters $\phi_1,...,\phi_k$ are exactly those first $k$ eigenvectors corresponding to the largest eigenvalues. The spectral filtering class is further parameterized by matrices $M_1,...,M_k \in \mathbb{R}^{d_{\text{out}} \times d_{\text{in}}}$. The output at time $t$ is then given by equation equation 2.

The following theorem establishes that the spectral filtering class of predictors approximately contains bounded linear dynamical systems with positive semi-definite $A$. The exact constants are left out for simplicity of presentation, but appear in the original work.

**Theorem 4** (Simplified from Hazan et al. (2017a)). *Given any linear dynamical system parametrized by $A, B, C, D$ such that $A$ is a PSD matrix with $\|A\| \leq 1$, there exists matrices $M_1,...,M_k$, such that for all $T$ and all sequences $u_{1:T}, \|u_t\| \leq 1$, the following holds. Let $y_{1:T}^{\text{LDS}}$ be*

the sequence generated by execution of the LDS via equation 1 and $y_{1:T}^{\mathrm{SF}}$ be the sequence generated by Spectral Filtering via equation 2. Then for all $t \in [T]$,

$$\|y_t^{\mathrm{LDS}} - y_t^{\mathrm{SF}}\| \sim e^{-\frac{k}{\log(L)}}.$$

Theorem 4 establishes that Spectral Filtering can predict long memory sequences since the statements holds even over marginally stable linear dynamical systems.

# 3  LEARNING WITH A SHORT CONTEXT—PROVABLE LENGTH GENERALIZATION FOR LINEAR DYNAMICAL SYSTEMS

In Algorithm 1, we modify the classical online learning algorithm for spectral filtering to use a shorter context window. To properly define our notion of length generalization, we need to distinguish between context lengths. Thus we introduce the notation for the loss observed with a context length $L$: letting $\hat{y}(M, L)$ denotes the prediction of $y_t$ using $M = [M_1, \ldots, M_k]$ and context window size $L$ as in Eq. 4 of Algorithm 1 we have

$$\ell_t(M, L) \stackrel{\text{def}}{=} \|\hat{y}(M, L) - y_t\|^2.$$

Note that this is overloaded notation compared with $\ell_t(y, y')$ which measures the loss of the true $y$ with the predicted $y'$ as used in our definition of regret. To provide a precise statement on length

---

**Algorithm 1** Spectral Filtering with Limited Context

---

1: **Input:** $k > 0, T > 0, L > 0, r > 0$. Initialize $M_i^1 \in \mathcal{R}^{d_{\mathrm{out}} \times d_{\mathrm{in}}}$ for $i \in [k]$ and set $M^1 = [M_1^1, \ldots, M_k^1]$. Let $\phi_{1:k}$ be the largest eigenvectors of $H_T$ defined in Eq. 3 with corresponding eigenvalues $\sigma_{1:k}$, and let $\pi_{\mathcal{K}}(\cdot)$ denote the projection to convex set $\mathcal{K}$.
2: **for** $t = 1, 2, ..., T$ **do**
3:     Compute and predict

$$\hat{y}_t = y_{t-1} + \sum\nolimits_{i=1}^{k} M_i^t u_{(t-1):(t-L)}(\sigma_i^{1/4}\phi_i). \tag{4}$$

4:     Observe $y_t$, denote $\ell_t(M^t, L) = \|\hat{y}_t - y_t\|^2$ and update and project onto the low Frobenius norm ball

$$\hat{M}^{t+1} \leftarrow M^t - \eta_t \nabla_M \ell_t(M^t)$$
$$M^{t+1} = \pi_{\mathcal{K}}\left(\hat{M}^{t+1}\right),$$

    where $\mathcal{K}_r = \left\{M \in \mathbb{R}^{k \times d_{\mathrm{out}} \times d_{\mathrm{in}}} \text{ s.t. } \|M_i\| \leq r \text{ for all } i \in [k]\right\}$.
5: **end for**

---

generalization, we present the following performance guarantee. Note that we prove the following for a $(A, B, C, I)$-LDS rather than $(A, B, C, D)$ which is without loss of generality since we can consider the input as $Du_1, \ldots, Du_T$.

**Theorem 5.** *Let $T \in \mathbb{Z}_{\geq 0}$ and $q \in [0, 1]$. Consider a sequence $(y_1, \ldots, y_T)$ generated by an unknown and noiseless linear dynamical system defined by matrices $(A, B, C, I)$ as per Eq. 1. Assume the input sequence $u_{0:(t-1)}$ is sufficiently well-conditioned, satisfying $\sum_{t=0}^{T-1}(T - t)u_t u_t^\top \succeq \left(\frac{2|C|\|B\|}{\sqrt{T}}\right) I$. Suppose the eigenvalues of $A$ lie within the range $\left[0, 1 - \frac{\log(T)}{8T^q}\right] \cup \left[1 - \frac{1}{2T^{5/4}}, 1\right]$. Let $k = \Omega\left(\log(T) \cdot \log(Td_A)\right)$, $r \geq \|B\|\|C\|$, and assume $T \geq (4k\log(T)/\|C\|\|B\|)^4$. Algorithm 1 satisfies:*

$$\mathit{Regret}_{\mathrm{Asymmetric},T}\left(\mathcal{A}(T^q), \Pi_T^{\mathrm{SF}}\right) \leq O\left(\|B\|^2\|C\|^2 k^{3/2} \log(T)\sqrt{T}\right).$$

The proof of Theorem 5 is in Appendix B with a high-level overview at the end of this section. This theorem shows that the sequence $M^1, \ldots, M^T$ constructed by Algorithm 1, even when using a reduced context length of size $T^q$, is able to achieve regret $O(\sqrt{T})$ when compared to the best

spectral filter that uses full context length $T$. To gain better understanding of the needed assumption on the spectrum of $A$, first suppose that all the eigenvalues of A are bounded by $1 - \delta$. Then the extent to which the input $u_{t-t_0}$ affects the value of $y_t$ is roughly $(1 - \delta)^{t_0}$, since the hidden state is multiplied by $A$ $t_0$ many times. This becomes negligible when $t_0$ is much larger than $1/\delta$ and implies that $u_{t-t_0}$ may be forgotten. This intuition explains why length generalization is possible for the first region of eigenvalues $[0, 1 - \log(T)/(8T^q)]$. Indeed, letting $\delta = \log(T)/8T^q$ and $t_0 = T^q$ (which is much bigger than $8T^q/\log(T)$ for large enough $T$) we see that when the spectrum of $A$ is smaller than $1 - \delta$, after $t_0$ many steps we can forget about the previous inputs $u_{t-t_0}$. The second part of the range – i.e. that the spectrum of A can lie between $[1 - 1/(2T^{5/4}), 1]$ – is a special feature of spectral filtering's ability to efficiently capture long memory effects and is rather technical. The "bad region" is exactly the range where the eigenvalues aren't small enough that $u_{t-t_0}$ can be forgotten for $t_0 \geq T^q$, but also aren't large enough that spectral filtering is naturally able to capture them. Numerically, the range is very small for large $T$ and reasonable $q$.

Motivated by the limitations of Theorem 5, in order to provide a length generalization that is robust to the spectrum of $A$, we introduce a variation on the classical Spectral Filtering algorithm, presented as Algorithm 2. This algorithm uses the two most previous outputs $y_{t-1}$ and $y_{t-2}$ when making a prediction $\hat{y}_t$ of $y_t$.

This algorithm has a slightly different construction of spectral filters. Indeed, they are the eigenvectors of the following matrix

$$N_T \stackrel{\text{def}}{=} \int_0^1 \tilde{\mu}_{\alpha,T} \tilde{\mu}_{\alpha,T}^\top d\alpha, \tag{5}$$

where $\tilde{\mu}_{\alpha,T} \stackrel{\text{def}}{=} (1-\alpha)^2[1, \alpha, \alpha^2, \ldots, \alpha^T]$. Interestingly, just by using one extra autoregressive term, our adapted algorithm is able to enjoy *robust* length generalization in the sense that whenever the context window is at least $T^{1/4+\epsilon}$ then no extra assumptions on the spectrum of $A$ are necessary to achieve our notion of length generalization. We state this formally in the following theorem.

---

**Algorithm 2** Spectral Filtering with Limited Context and Two Autogressive Components
___

1: **Input:** $k > 0, T > 0, L > 0, r > 0$. Initialize $M_i^1 \in \mathcal{R}^{d_{\text{out}} \times d_{\text{in}}}$ for $i \in [k]$ and set $M^1 = [M_1^1, \ldots, M_k^1]$. Let $\tilde{\phi}_{1:k}$ be the largest eigenvectors of $N_{T-2}$ defined in Eq. 5 with corresponding eigenvalues $\tilde{\sigma}_{1:k}$, and let $\pi_\mathcal{K}(\cdot)$ denote the projection to convex set $\mathcal{K}$.

2: **for** $t = 1, 2, ..., T$ **do**

3:    Compute and predict

$$\hat{y}_t = 2y_{t-1} - y_{t-2} + M_1^t u_{t-1} + M_2^t u_{t-2} + \sum_{i=3}^k M_i^t u_{(t-3):(t-L)}(\tilde{\sigma}_i^{1/4} \tilde{\phi}_i).$$

4:    Observe $y_t$, denote $\ell_t(M^t, L) = \|\hat{y}_t - y_t\|^2$ and update and project onto the low Frobenius norm ball

$$\hat{M}^{t+1} \leftarrow M^t - \eta_t \nabla_M \ell_t(M^t)$$

$$M^{t+1} = \pi_\mathcal{K}\left(\hat{M}^{t+1}\right),$$

   where $\mathcal{K}_r = \{M = [M_1, \ldots, M_k] \text{ s.t. } \|M_i\| \leq r \text{ for all } i \in [k]\}$.

5: **end for**

---

**Theorem 6.** *Let* $T \in \mathbb{Z}_{\geq 0}$ *and* $q \in \left[\frac{1}{4} + \frac{\log(\log(T)/8)}{\log(T)}, 1\right]$. *Consider a sequence* $(y_1, \ldots, y_T)$ *generated by an unknown and noiseless linear dynamical system defined by matrices* $(A, B, C, I)$ *as per Eq. 1. Assume the input sequence* $u_{0:(t-1)}$ *is sufficiently well-conditioned, satisfying* $\sum_{t=0}^{T-1}(T-t)u_t u_t^\top \succeq \left(\frac{2|C||B|}{\sqrt{T}}\right) I$. *Let* $k = \Omega\left(\log(T) \cdot \log(Td_A)\right)$, $r \geq \|B\|\|C\|$ *and assume* $T \geq (4k\log^2(T)/\|C\|\|B\|)^4$. *Algorithm 2 satisfies:*

$$Regret_{\text{Asymmetric},T}\left(\mathcal{A}(T^q), \Pi_T^{\text{SF}}\right) \leq O\left(\|B\|^2\|C\|^2 k^{3/2} \log^2(T)\sqrt{T}\right).$$

The proof of Theorem 6 is in Appendix C and we now give a high-level overview.

**High-Level Proof Overview.** The general proof technique for both Theorem 5 and Theorem 6 is the same. First, using standard online gradient descent results from Hazan et al. (2017b) we prove that the iterates $M^t$ achieve $O(\sqrt{T})$ regret as measured by the context-length restricted loss $\sum_{t=1}^{T} \ell_t(M, L)$. That is,

$$\sum_{t=1}^{T} \ell_t(M^t, L) \leq \min_{M \in \mathcal{K}_r} \sum_{t=1}^{T} \ell_t(M, L) + O(\sqrt{T}). \tag{6}$$

Next we prove that there is a unique $M_T^*$ which minimizes the loss on the full $T$-length context and this $M_T^*$ achieves length generalization in the sense that it achieves small loss even when only allowed to use context length $L$. That is

$$\sum_{t=1}^{T} \ell_t(M_T^*, L) \leq \sum_{t=1}^{T} \ell_t(M_T^*, T) + O(\sqrt{T}). \tag{7}$$

We combine Eq. 6 and Eq. 7 to get the final notion of length generalization that

$$\sum_{t=1}^{T} \ell_t(M^t, L) \leq \min_{M \in \mathcal{K}_r} \sum_{t=1}^{T} \ell_t(M, L) + O(\sqrt{T}) \leq \sum_{t=1}^{T} \ell_t(M_T^*, L) + O(\sqrt{T}) \leq \sum_{t=1}^{T} \ell_t(M_T^*, T) + O(\sqrt{T}).$$

The difficult result to prove is Eq. 7. The high level idea is that when $y_{1:t}$ evolves as a noiseless LDS and when the input $u_{0:(t-1)}$ is sufficiently well-conditioned, then $\sum_{t=1}^{T} \ell_t(M, T)$ is strongly convex and the minimizer approximately recovers a collection of "true" matrices which are generated by the underlying linear dynamical system. The second key idea is that if an algorithm had access to these "true" matrices then it would be able to achieve small loss even when restricted to a small context-length $L \ll T$. The extent to which these recovered matrices can achieve small loss when restricted to the small context-length depends on the way the algorithm chooses to predict $y_t$. In the case of Algorithm 1 where $y_t$ is predicted based only using only one autoregressive term, even having access to the true matrices is not enough to accurately predict $y_t$. However, in the case of Algorithm 2, having access to the true matrices as well as a second autoregressive term allows accurate prediction of $y_t$ even when restricted to small context-length window.

## 4 EXPERIMENTS

### 4.1 LINEAR DYNAMICAL SYSTEM

We can empirically verify Theorem 5 and Theorem 6 in an online sequence prediction task where the data is generated by a noiseless LDS. We refer to a "bad" region of eigenvalues $\left(1 - \log(T)/(8T^{7/8}),\ 1 - 1/(2T^{5/4})\right)$ as Region B, and we define Region A to hug Region B on both sides as shown in Figure 1. Theorem 5 predicts that if all the eigenvalues lie outside Region B,

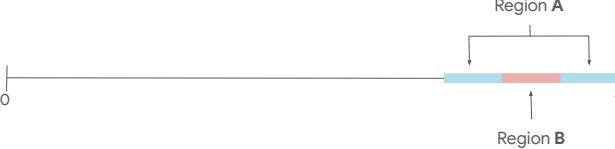

Figure 1: Region B is the interval of eigenvalues for which Theorem 5 does not provide length generalization. Region A hugs both sides of Region B (Region A is $[0.9 \cdot \left(1 - \log(T)/(8T^{7/8})\right), 1] \setminus$ Region B. This ensures that Region $A$ will contain bad eigenvalues as $q$ decreases from $7/8$ and eigenvalues in Region B are bad for $q \leq 7/8$.

then spectral filtering will length generalize from $T^{7/8}$ to $T$. To confirm this, we generate a random LDS (hidden dimension: 512) with half of its eigenvalues sampled from each part of **Region A**. The online prediction losses are plotted in Figure 2 for different choices of context length $T^q$, where $T = 2^{14}$ and $k = 24$. As expected from the theory, context lengths approaching $T^{7/8}$ closely match the performance of the optimal spectral filtering predictor with full context.

Interestingly, we see that context length $T^{1/2}$ consistently fails in a qualitatively worse fashion – indeed, some of the values in Region A are actually "bad" for $q = 1/2$. This seems to suggest that

such eigenvalues can actually cause instabilities with length generalization and are not limitations of our proof – if true, such a fact could be seen as a partial converse to Theorem 5. To check this conjecture empirically, we run another experiment where we generate a random LDS of hidden dimension 512 with all eigenvalues in **Region B** and plot the prediction losses $\ell_t(M^t, T^q)$ for $M^t$ from Algorithm 1 in Figure 3 (averaged over random seeds and smoothed). These results confirm that (some subset of) this bad region is indeed what impedes the length generalization capability of spectral filtering.

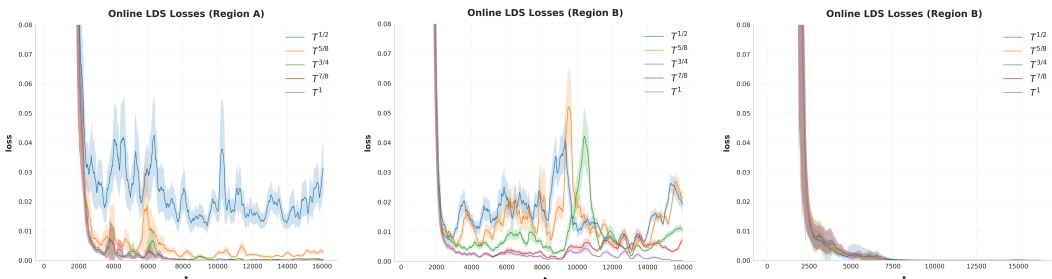

Figure 2: Loss for Algorithm 1 with eigenvalues in **Region A**.

Figure 3: Loss for Algorithm 1 with eigenvalues in **Region B**.

Figure 4: Loss for Algorithm 2 with eigenvalues in **Region B**.

Next we apply our novel Algorithm 2, which uses two autoregressive components. Theorem 6 predicts that this algorithm should be robust to this bad region of eigenvalues and instead achieve length generalazation for any (symmetric, marginally-stable) LDS. We verify this experimentally in Figure 4 – to be as adversarial as we can, this experiment is run with all eigenvalues sampled from **Region B**. As predicted by Theorem 6, the second autoregressive component allows for robust length generalization even with context lengths as small as $\sqrt{T}$.

## 4.2 INDUCTION HEADS

So far, we have demonstrated length generalization of spectral filtering on linear systems: when trained with a shorter context length of $T^q$ it is able to compete with methods that have access to the full context $T$ (even on marginally-stable systems that can have arbitrarily large effective memory lengths). This length generalization property is most crucial in deep learning applications, in which multi-layer models are stacked (with added nonlinearities) to solve non-LDS sequence prediction tasks.

As an empirical proof-of-concept to demonstrate that STU's length generalization capability extends to this regime, we evaluate it on the induction heads synthetic sequence modeling task, which is commonplace in the language modeling literature (see Gu & Dao (2023)) and was experimentally shown in Liu et al. (2024) to be efficiently solved by a two-layer STU. In the induction heads task, the model is required to recall one token (sampled uniformly from a vocabulary) immediately after a special `flag` token; the rest of the sequence consists of the same special `blank` token, which the model should learn to ignore.

The STU architecture we use is composed of an embedding layer, two "tensordot" STU layers with MLPs and ReLU nonlinearities, and an output projection layer (the same as in Liu et al. (2024)) with filters of length $T = 256$.

Following prior STU architecture implementations we use **no autoregressive components**, and so any length generalization observed here comes directly from the filtering mechanism itself. We train these models until convergence with a tuned Adam optimizer and various context lengths $T^q$. The vocabulary size is set to 4.

Accuracies are plotted in Figure 5 for evaluation task lengths increasing up to $T$. As we see, vanilla STU models are able to nontrivially length generalize and occasionally retain good accuracy beyond

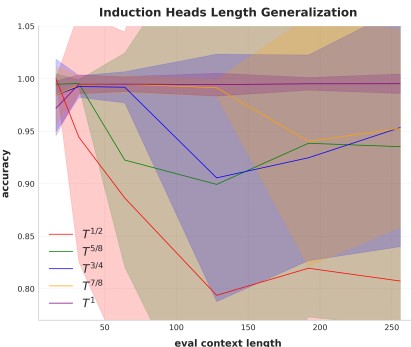

Figure 5: Accuracies for STU models trained on an induction heads task of length $T^q$ and evaluated on sequence lengths increasing up to $T$, averaged over random seeds. Models occasionally generalize all the way up to length $T$, as indicated by the large variance of evaluation accuracies.

their training context lengths, though inconsistently[2]. Importantly, unlike algorithms that achieve length generalization through architectural modification, we simply just train with filters longer than the train context. As such, this method allows for the convolutional mode during training and inherits all the benefits of STU that are demonstrated in Liu et al. (2024). For example, the nonlinear selection mechanism of Gu & Dao (2023) allows for extreme length generalization on induction heads without prior knowledge of the evaluation length, though at a cost to training efficiency, implementation simplicity, and optimization complexity. We reiterate that our goal is not to navigate such a tradeoff by modifying the STU model so that it length generalizes on induction heads, but rather to exhibit a provable length generalization capability of the STU that comes for free from its natural structure.

## 5 DISCUSSION

In review, we first introduced the notion of Asymmetric-Regret as a way to describe length generalization through the lens of online learning and regret minimization in games. We then proved that the class of spectral filtering predictors naturally enjoys sublinear Asymmetric-Regret thereby exhibiting length generalization without any change to the algorithm, albeit with some restrictions on the underlying data (i.e. the spectrum of $A$). We introduced a new variant of spectral filtering which uses two autoregressive components and achieves length generalization which is more robust to the assumptions of the underlying data. Next, we used experiments on synthetic data generated by an LDS to demonstrate the validity and sharpness of our theory and provided proof-of-concept length generalization experiments on a synthetic nonlinear sequence prediction task.

Our theoretical results and initial empirical findings reveal that some type of length generalization comes naturally with the spectral filtering algorithm. This result implies that spectral filtering is powerful in its ability to learn the dynamics of a complicated underlying system with long memory – it naturally handles the issue of what aspects in a sequence should be memorized for the future and what aspects can be forgotten, whereas many existing methods are hand engineered depending on the specific task. This adds to the already-exciting list of its useful (and provable) properties, including: robustness to systems with long memory and large hidden dimension, efficient training via convolutions, optimization convexity, and the existence of good parameter-efficient approximations. Given recent successful applications of spectral filtering as the building block for STU models in deep learning (Agarwal et al., 2023; Liu et al., 2024), it would be valuable to research how to best take advantage of their length generalization capacity at scale – we leave this for future work.

---

[2]The large variance in Figure 5 is due to bimodality in the accuracies – often the model generalizes perfectly, though sometimes it fails. Overcoming this through regularization or optimization considerations is a modeling question that ought to be studied in large empirical setups. We use this synthetic task strictly as a proof-of-concept: length generalization in synthetic tasks can be very sensitive (compare Figures 5 and 6 in Jelassi et al. (2024), for example), and it can be difficult to know when length generalization on a certain task informs us about real-world applications Ben-Kish et al. (2024). We leave a thorough empirical study on length generalization in LLMs to future work.

