# A  GENERAL LENGTH GENERALIZATION

In this section we introduce a general algorithm which we will use to prove length generalization for both Algorithm 1 and Algorithm 2.

---

**Algorithm 3** General Spectral Filtering

---

1: **Input:** $k > 0, L > 0, r > 0$, functions $p_t(\cdot)$, vectors $v_{1:k}$. Initialize $M_i = 0$ for $i \in [k]$.
2: **for** $t = 1, 2, ..., T$ **do**
3:   Compute and predict

$$\hat{y}_t = p_t(y_{t-1:1}) + \sum_{i=1}^{k} M_i u_{t-1:t-L} v_i.$$

4:   Observe $y_t$, denote $\ell_t(M^t, L) = \|\hat{y}_t - y_t\|^2$ and update and project update and project onto the low Frobenius norm ball

$$\hat{M}^{t+1} \leftarrow M^t - \eta_t \nabla_M \ell_t(M^t)$$
$$M_{t+1} = \text{Proj}_{\mathcal{K}} \left( \hat{M}_{t+1} \right),$$

   where $\mathcal{K}_r = \{M \text{ s.t. } \|M_i\| \leq r\}$.
5: **end for**

---

Our workhorse theorem is presented below. We will use this theorem to prove length generalization for our special cases in the following sections.

**Theorem 7.** *Suppose $y_{1:t}$ evolves as a noiseless $(A, B, C, I)$-LDS and the input $u_{(t-1):0}$ is such that $\sum_{t=0}^{T-1} (T - t) u_t u_t^\top \succeq (2\|C\|\|B\|/\sqrt{T})I$. Let $k$, $L$, $r$, $\{v_i\}_{i=1}^{k}$, $p_t(\cdot)$, and $\ell_t(\cdot)$ all be as defined in Algorithm 3. Suppose $\{v_i\}_{i=1}^{k}$ is orthonormal with $\|v_i\|_1 \leq \log^p(T)$. Suppose that $p_t(\cdot)$ is such that there exists some function $h(\cdot)$, constant $\ell > 0$, and some $M^{true} \in \mathcal{K}_r$ such that*

$$y_t - p_t(y_{t-1:1}) = \sum_{i=1}^{T} M_i^{true} u_{t-1:0} v_i = \sum_{i=1}^{\ell_1} M_i^{true} u_{t-i} + \sum_{i=1}^{t-\ell_1} C A^i h(A) B u_{t-\ell_1-i},$$

*where*

$$\| \sum_{i=k+1}^{T} M_i^{true} u_{t-1:t-L} v_i \| \leq \|C\|\|B\|/T,$$

*and*

$$\max_{\alpha(A)} \left\{ h(\alpha) \alpha^{L-\ell_1-1} (1 - \alpha^{T-L+1})(1 - \alpha)^{-1} \right\} \leq \frac{1}{T^{1/4}}.$$

*Then if $M^t$ are the iterates of Algorithm 3 and $T \geq (4k \log^p(T)/\|C\|\|B\|)^4$,*

$$\sum_{t=1}^{T} \ell_t(M^t, L) - \min_{M^* \in \mathcal{K}_r} \sum_{t=1}^{T} \ell_t(M^*, T) \leq \left( 12 k^{3/2} r^2 \log^p(T) + 8\|C\|^2\|B\|^2 \right) \sqrt{T}.$$

The proof of this theorem requires several technical lemmas which we present and prove in the subsequent subsections. In Lemma 8 we essentially prove the standard result showing that Online Gradient Descent implemented in Algorithm 3 achieves $O(\sqrt{T})$ regret. In Lemma 9 we prove the more nuanced result which shows that the optimal $M$ which minimizes the loss on the full $T$-length context achieves length generalization in the sense that it achieves small loss even when only allowed to use context length $L$. Combining these two lemmas gives the proof of Theorem 7.

*Proof of Theorem 7.* Let

$$M_T^* \overset{\text{def}}{=} \min_{M^* \in \mathcal{K}_r} \sum_{t=1}^{T} \ell_t(M^*, T)$$

and observe that

$$\min_{M^* \in \mathcal{K}_r} \sum_{t=1}^{T} \ell_t(M^*, L) \ \leq \ \sum_{t=1}^{T} \ell_t(M_T^*, L). \qquad (8)$$

Combining this with Lemma 8 and Lemma 9, we conclude

$$\sum_{t=1}^{T} \ell_t(M_t, L) \ \leq \ \min_{M^* \in \mathcal{K}_r} \sum_{t=1}^{T} \ell_t(M^*, L) + 12k^{3/2} r^2 \log^p(T) \sqrt{T} \qquad \text{OGD Regret Lemma 8}$$

$$\leq \ \sum_{t=1}^{T} \ell_t(M_T^*, L) + 12k^{3/2} r^2 \log^p(T) \sqrt{T} \qquad \text{Eq. 8}$$

$$\leq \ \sum_{t=1}^{T} \ell_t(M_T^*, T) + (12k^{3/2} r^2 \log^p(T) + 8\|C\|^2 \|B\|^2) \sqrt{T} \qquad \text{Length Generalization Lemma 9}$$

$$= \ \min_{M^* \in \mathcal{K}_r} \sum_{t=1}^{T} \ell_t(M, T) + (12k^{3/2} r^2 \log^p(T) + 8\|C\|^2 \|B\|^2) \sqrt{T}. \qquad \text{Definition of } M_T^*$$

$$\square$$

### A.1 OGD Regret for Generalized Spectral Filtering

**Lemma 8.** *Suppose the input $u_{1:t}$ satisfies $\|u_t\|_2 \ \leq \ 1$. Suppose the true output $y_t$ evolves such that for some polynomial $p_t(y_{t-1:1})$ there exists some $M^{true} \in \mathcal{K}_r$*

$$y_t = p_t(y_{t-1:1}) + \sum_{i=1}^{T} M_i^{true} u_{t-1:0} v_i,$$

*and for*

$$E_{m,T} \ \stackrel{def}{=} \ \sum_{i=k+1}^{T} M_i^{true} u_{t-1:0} v_i,$$

*we have $\|E_{m,T}\| \ \leq \ 1$. Further suppose $v_1, \ldots, v_k$ satisfy $\|v_i\|_1 \ \leq \ c_i \log^p(T)$. Let*

$$\ell_t(M, L) \ \stackrel{def}{=} \ \|y_t - p_t(y_{t-1:1}) - \sum_{i=1}^{k} M_i u_{t-1:t-L} v_i\|^2,$$

*Then if $M^t$ are the iterates of Algorithm 3*

$$\sum_{t=1}^{T} \ell_t(M^t, L) - \min_{M^* \in \mathcal{K}_r} \sum_{t=1}^{T} \ell_t(M^*, L) \ \leq \ 12k^{3/2} r^2 \log^p(T) \sqrt{T}.$$

*Proof of Lemma 8.* This proof is a near copy of the proof in Hazan et al. (2017b), the difference is that we derive several equations that we will use later and we handle the varying context length.

Let $G = \max_{t \in [T]} \|\nabla_M \ell_t(M_t, L)\|$ and let $D = \max_{M_1, M_2 \in \mathcal{K}_r} \|M_1 - M_2\|$. By Theorem A.1 from Hazan & Singh (2022),

$$\sum_{t=1}^{T} \ell_t(M^t, L) - \min_{M^* \in \mathcal{K}_r} \sum_{t=1}^{T} \ell_t(M^*, L) \ \leq \ \frac{3}{2} GD\sqrt{T}.$$

Therefore it remains to bound $G$ and $D$.

First we bound $D$. By definition of $\mathcal{K}_r$, we have that for any $M \in \mathcal{K}_r$,

$$\|M_i\| \ \leq \ r.$$

Therefore, we also have that
$$\|M\| \leq \sqrt{k}r.$$

Therefore
$$D \overset{\text{def}}{=} \max_{M,M'\in\mathcal{K}_r} \|M-M'\| \leq 2\sqrt{k}r.$$

Next we bound the gradient norm $G$. Using the definition of $\mathcal{K}_r$,
$$\max_{M\in\mathcal{K}_r} \max_{i\in[k]} \|M_i\| \leq r.$$

We bound the gradient norm as follows,

$$\|\nabla_{M_j}\ell_t(M,L)\| = \|2\left(\sum_{i=1}^{k} M_i^{\text{true}}u_{t-1:0}v_i + E_{m,T} - \sum_{i=1}^{k} M_i u_{t-1:t-L}v_i\right)(u_{t-1:t-L}v_j)^\top\|$$

$$\leq 2\left(\sum_{i=1}^{k} \|M_i^{\text{true}}\|\|u_{t-1:0}\|_\infty\|v_i\|_1 + \|E_{m,T}\| + \sum_{i=1}^{k}\|M_i\|\|u_{t-1:t-L}\|_\infty\|v_i\|_1\right)\|u_{t:t-L}\|_\infty\|v_j\|_1$$

$$\leq 2\left(1 + \|E_{m,T}\|\right)\sum_{i=1}^{k} \max_{M\in\mathcal{K}_r} \|M_i\| \cdot \|u_{t-1:0}\|_\infty^2 \cdot \|v_i\|_1^2$$

$$\leq 4kr\log^p(T).$$

Putting everything together we have

$$\sum_{t=1}^{T}\ell_t(M_t,L) - \min_{M^*\in\mathcal{K}_r}\sum_{t=1}^{T}\ell_t(M^*,L) \leq \frac{3}{2}\left(4kr\log^p(T)\right)\left(2\sqrt{k}r\right)\sqrt{T}$$

$$= 12k^{3/2}r^2\log^p(T)\sqrt{T}.$$

$\square$

## A.2 Length Generalization on the Best Optimizer in Hindsight

**Lemma 9.** *Let input $u_{(t-1):0}$, $\{v_i\}_{i=1}^{k}$, $p_t(\cdot)$, and $\ell_t(M,L)$ all be as defined in Algorithm 3. Suppose the input $u_{(t-1):0}$ is such that $\sum_{t=0}^{T-1}(T-t)u_t u_t^\top \succeq (2\|C\|\|B\|/\sqrt{T})I$, $\{v_i\}_{i=1}^{k}$ is orthonormal with $\|v_i\|_1 \leq \log^p(T)$, and that there exists some $M^{true}$ such that*

$$y_t - p_t(y_{t-1:1}) = \sum_{i=1}^{T} M_i^{true}u_{t-1:0}v_i = \sum_{i=1}^{\ell_1} M_i^{true}u_{t-i} + \sum_{i=1}^{t-\ell_1-1} CA^i h(A)Bu_{t-\ell_1-i},$$

*where*

$$\|\sum_{i=k+1}^{T} M_i^{true}u_{t-1:t-L}v_i\| \leq \|C\|\|B\|/T,$$

*and*

$$\max_{\alpha(A)}\left\{h(\alpha)\alpha^{L-\ell_1-1}(1-\alpha^{T-L+1})(1-\alpha)^{-1}\right\} \leq \frac{1}{T^{1/4}}.$$

*Let*

$$M_T^* \overset{def}{=} \arg\min_{M\in\mathcal{K}_r}\sum_{t=1}^{T}\ell_t(M,T).$$

*Then for $T \geq (4k\log^p(T)/\|C\|\|B\|)^4$, the loss with context $L$ well approximates the loss with context $T$ on $M_T^*$,*

$$|\sum_{t=1}^{T}\ell_t(M_T^*,L) - \ell_t(M_T^*,T)| \leq 8\|C\|^2\|B\|^2\sqrt{T}.$$

The proof of Lemma 9 requires two key helper lemmas which we develop in the following subsections. The first is Lemma 10 which establishes that when $y_{1:t}$ evolves as a noiseless LDS and if the input $u_{1:t}$ is sufficiently well-conditioned, then the minimizer for $\sum_{t=1}^{T} \ell_t(M, T)$ approximately recovers a collection of matrices (we denote as $M^{\text{true}}$) which is generated by the true linear dynamical system. The second key helper Lemma is Lemma 11 which establishes that an algorithm which uses the collection of matrices that are generated by the true linear dynamical system, i.e. $M^{\text{true}}$, is able to achieve small loss even when restricted to a small context-length $L << T$. The proof of Lemma 9 combines these two insights to establish that this implies that the minimizer for $\sum_{t=1}^{T} \ell_t(M, T)$ also achieves small loss even when restricted to small context-length $L$.

*Proof of Lemma 9.* First we show that $M^{\text{true}}$ is a $(\|C\|^2 \|B\|^2 / T)$-approximate minimizer to $\sum_{t=1}^{T} \ell_t(M, T)$. Indeed,

$$\sum_{t=1}^{T} \ell_t(M^{\text{true}}, T) = \sum_{t=1}^{T} \| y_t - p_t(y_{t-1:1}) - \sum_{i=1}^{k} M_i^{\text{true}} u_{t-1:0} v_i \|^2$$

$$= \sum_{t=1}^{T} \| \sum_{i=k+1}^{T} M_i^{\text{true}} u_{t-1:0} v_i \|^2$$

$$\leq \|C\|^2 \|B\|^2 / T.$$

By assumption $\sum_{t=0}^{T-1} (T-t) u_t u_t^\top \succeq (2\|C\|\|B\|/\sqrt{T}) I$. Therefore, by Lemma 10 with $\epsilon = \|C\|\|B\|/\sqrt{T}$ we have

$$M_T^* \in \mathcal{B}_{\|C\|\|B\|/\sqrt{T}} \left( M^{\text{true}} \right).$$

Since we assumed $T \geq (4k \log^p(T)/\|C\|\|B\|)^4$ we have

$$\|C\|\|B\|/\sqrt{T} \leq \|C\|^2 \|B\|^2 / (4k T^{1/4} \log^p(T)).$$

Therefore by Lemma 11 we have

$$\sum_{t=1}^{T} \ell_t(M_T^*, L) \leq 4\|C\|^2 \|B\|^2 \sqrt{T}.$$

Moreover note that

$$0 \leq \ell_t(M_T^*, T) \leq \ell_t(M^{\text{true}}, T) \leq \|C\|^2 \|B\|^2 / T^2.$$

Combining these we conclude,

$$|\sum_{t=1}^{T} \ell_t(M_T^*, L) - \sum_{t=1}^{T} \ell_t(M_T^*, T)| \leq 4\|C\|^2 \|B\|^2 \sqrt{T} + \|C\|^2 \|B\|^2 / T \leq 8\|C\|^2 \|B\|^2 \sqrt{T}.$$

$\square$

### A.2.1 MINIMIZATION IS RECOVERY

**Lemma 10.** *Suppose $\sum_{t=0}^{T-1} (T-t) u_t u_t^\top \succeq 2\epsilon I$ and $\{v_i\}_{i=1}^k$ is orthonormal. Then there is a unique point $M^*$ which minimizes the function $\sum_{t=1}^{T} \ell_t(M, T)$ from Algorithm 3. Moreover, suppose some $k$ satisfies*

$$\sum_{t=1}^{T} \ell_t(M, T) \leq \epsilon^2.$$

*Then there is a matrix $E_M$ such that $\|E_M\| \leq \epsilon$ and*

$$M^* = M + E_M.$$

*Proof.* For convenience, let $X_t$ be the $kd_{\text{in}}$-dimensional vector which stacks the filters,

$$X_t = \begin{bmatrix} u_{t-1:t-T} v_1 \\ u_{t-1:t-T} v_2 \\ \vdots \\ u_{t-1:t-T} v_k \end{bmatrix} = \begin{bmatrix} u_{t-1:0} v_1 \\ u_{t-1:0} v_2 \\ \vdots \\ u_{t-1:0} v_k \end{bmatrix},$$

where the second inequality holds since we only consider $t \leq T$. Assume $k$ is written as $M = [M_1 \quad M_2 \quad \ldots \quad M_k] \in \mathbb{R}^{d_{\text{out}} \times k d_{\text{in}}}$ and let $Y_t = y_t - p_t(y_{t-1:1})$. Let $Y = [Y_1 \quad Y_2 \quad \ldots \quad Y_T]$ and $X = [X_1 \quad X_2 \quad \ldots \quad X_T]$. Then we can express the loss as

$$f(M) \overset{\text{def}}{=} \sum_{t=1}^{T} \ell_t(M, T) = \|Y - MX\|^2.$$

Note that this function is twice differentiable and

$$\nabla_M^2 f(M) = XX^\top.$$

Therefore, if $\lambda_{\min}\left(XX^\top\right) \geq \mu$ we have that $f(M)$ is $\mu$-strongly convex. Then if $M^*$ is the optimum of $f(M)$ we have

$$f(M) \geq f(M^*) + \frac{\mu}{2}\|M - M^*\|^2, \text{ or equivalently, } \|M - M^*\| \leq \frac{2}{\mu}\left(f(M) - f(M^*)\right).$$

Now suppose $k$ is such that $f(M) \leq \epsilon^2$. Then since $f(\cdot) \geq 0$ we have

$$\|M - M^*\| \leq 2\epsilon^2/\mu.$$

Therefore we can write

$$M^* = M + E_{M^*} \text{ where } \|E_{M^*}\| \leq 2\epsilon^2/\mu. \tag{9}$$

Next we must understand the eigenvalues of $XX^\top$ and how they relate to the input $u_{T:1}$. For notational convenience, let $U = u_{T:1}$ and let $D_t$ denote the block-diagonal $T \times T$ matrix

$$D_t \overset{\text{def}}{=} \begin{bmatrix} 0_{T-t \times T-t} & \\ & I_t \end{bmatrix}.$$

Finally, let

$$V = \begin{bmatrix} v_1 \\ v_2 \\ \vdots \\ v_k \end{bmatrix} \in \mathcal{R}^{Tm \times 1}$$

Then we have $X_t = (I_k \otimes UD_t) V$ and we observe

$$XX^\top = \sum_{t=1}^{T} X_t X_t^\top = \sum_{t=1}^{T} \left((I_k \otimes UD_t)V\right)\left((I_k \otimes UD_t)V\right)^\top$$

$$= \sum_{t=1}^{T} (I_k \otimes UD_tU^\top)$$

$$= I_k \otimes U\left(\sum_{t=1}^{T} D_t\right)U^\top.$$

Observe that

$$\sum_{t=1}^{T} D_t = \text{diag}\left([1 \quad 2 \quad \ldots \quad T]\right).$$

Using this we can further refine

$$U\left(\sum_{t=1}^{T} D_t\right)U^\top = \sum_{t=0}^{T-1}(T-t)u_t u_t^\top.$$

By assumption, this matrix has minimum eigenvalue bounded below by $2\epsilon$. Therefore $\lambda_{\min}(XX^\top) \geq 2\epsilon$. Plugging this value in for $\mu$ in Eq. 9 concludes the proof.

$\square$

## A.2.2 Uniform Length Generalization Around LDS Generated Solutions

The following lemma shows that any $k$ in an (appropriately defined) $\epsilon$-ball around $M^{\text{true}}$ obtains length generalization in the sense that it achieves $O(\sqrt{T})$ $L$-context-length-limited loss $\sum_{t=1}^{T} \ell_t(\cdot, L)$.

**Lemma 11.** *Suppose $y_t$ evolves as a noiseless $(A, B, C, I)$-LDS with input $u_t$. Suppose $p_t(\cdot)$ and $M^{\text{true}}$ is such that*

$$y_t - p_t(y_{t-1:1}) = \sum_{i=1}^{T} M_i^{\text{true}} u_{t-1:0} v_i = \sum_{i=1}^{\ell_1} M_i^{\text{true}} u_{t-i} + \sum_{i=1}^{t-\ell_1-1} CA^i h(A) B u_{t-\ell_1-i}.$$

*Suppose for a given $k > 0$,*

$$\| \sum_{i=k+1}^{T} M_i^{\text{true}} u_{t-1:t-L} v_i \| \leq \frac{\|C\|\|B\|}{T}.$$

*Suppose*

$$\max_{\alpha(A)} \left\{ h(\alpha) \alpha^{L-\ell_1-1} (1 - \alpha^{T-L+1})(1 - \alpha)^{-1} \right\} \leq \frac{1}{T^{1/4}}.$$

*If*

$$\delta \leq \frac{1}{4m} \frac{\|C\|^2 \|B\|^2}{T^{1/4} \log^p(T)},$$

*then we have for any $M \in \mathcal{B}_\delta(M^{\text{true}})$*

$$\sum_{t=1}^{T} \ell_t(M, L) \leq 4\|C\|^2 \|B\|^2 \sqrt{T}.$$

*Proof of Lemma 11.* Let $M = M^{\text{true}} + E_M$, where $\|E_M\| \leq \delta$. By definition,

$$\ell_t(M^{\text{true}} + E_M, L) = \|y_t - p_t(y_{t-1:1}) - \sum_{i=1}^{k} \left(M^{\text{true}} + E_M\right)_i u_{t-1:t-L} v_i \|^2$$

$$= \|y_t - p_t(y_{t-1:1}) - \sum_{i=1}^{k} M_i^{\text{true}} u_{t-1:t-L} v_i - \sum_{i=1}^{k} E_{M_i} u_{t-1:t-L} v_i \|^2$$

$$\leq \|y_t - p_t(y_{t-1:1}) - \sum_{i=1}^{k} M_i^{\text{true}} u_{t-1:t-L} v_i \|^2$$

$$+ 2\|y_t - p_t(y_{t-1:1}) - \sum_{i=1}^{k} M_i^{\text{true}} u_{t-1:t-L} v_i \| \| \sum_{i=1}^{k} E_{M_i} u_{t-1:t-L} v_i \|$$

$$+ \| \sum_{i=1}^{k} E_{M_i} u_{t-1:t-L} v_i \|^2.$$

Observe that

$$\| \sum_{i=1}^{k} E_{M_i} u_{t-1:t-L} v_i \| \leq \sum_{i=1}^{k} \|E_{M_i}\| \|u_{t-1:t-L}\|_\infty \|v_i\|_1 \leq k\delta \log^p(T).$$

For the remainder of the proof we work towards bounding $\|y_t - p_t(y_{t-1:1}) - \sum_{i=1}^{k} M_i^{\text{true}} u_{t-1:t-L} v_i\|$. We replace $y_t - p_t(y_{t-1:1})$ with $\sum_{i=1}^{T} M_i^{\text{true}} u_{t-1:0} v_i$ and we replace $\sum_{i=1}^{k} M_i^{\text{true}} u_{t-1:t-L} v_i$ with $\sum_{i=1}^{T} M_i^{\text{true}} u_{t-1:t-L} v_i -$

$\sum_{i=k+1}^{T} M_i^{\text{true}} u_{t-1:t-L} v_i$ to get

$$\|y_t - p_t(y_{-1:1}) - \sum_{i=1}^{k} M_i^{\text{true}} u_{t-1:t-L} v_i\|^2 = \|\left(\sum_{i=1}^{T} M_i^{\text{true}} u_{t-1:0} v_i\right) - \left(\sum_{i=1}^{T} M_i^{\text{true}} u_{t-1:t-L} v_i - \sum_{i=k+1}^{T} M_i^{\text{true}} u_{t-1:t-L} v_i\right)\|^2$$

$$\leq \|\sum_{i=1}^{T} M_i^{\text{true}}(u_{t-1:0} - u_{t-1:t-L}) v_i\|^2$$

$$+ 2\|\sum_{i=1}^{T} M_i^{\text{true}}(u_{t-1:0} - u_{t-1:t-L}) v_i\|\|\sum_{i=k+1}^{T} M_i^{\text{true}} u_{t-1:t-L} v_i\|$$

$$+ \|\sum_{i=k+1}^{T} M_i^{\text{true}} u_{t-1:t-L} v_i\|^2.$$

Next we note that $\|\sum_{i=k+1}^{T} M_i^{\text{true}} u_{t-1:t-L} v_i\|$ is assumed to be at most $\|C\|\|B\|/T$ and so we now focus on bounding the norm:

$$\|\sum_{i=1}^{T} M_i^{\text{true}}(u_{t-1:0} - u_{t-1:t-L}) v_i\|. \tag{10}$$

Towards bounding Eq. 10, assume $L > \ell_1$ so that

$$\sum_{i=1}^{T} M_i^{\text{true}}(u_{t-1:0} - u_{t-1:t-L}) v_i = \sum_{i=L-\ell_1+1}^{t-\ell_1-1} C A^i h(A) B u_{t-\ell_1-i}$$

$$= \sum_{i=L-\ell_1+1}^{t-\ell_1-1} \sum_{j=1}^{d_A} \alpha_j^i h(\alpha_j) C_j B_j^\top u_{t-\ell_1-i}.$$

Then

$$\|\sum_{i=L-\ell_1+1}^{t-\ell_1-1} C A^i h(A) B u_{t-\ell_1-i}\| \leq \max_{j \in [d_A]} \alpha_j^i h(\alpha_j) \sum_{i=L-\ell_1+1}^{t-\ell_1-1} \|C_j B_j^\top u_{t-\ell_1-i}\|$$

$$\leq \max_{\alpha(A)} \sum_{i=L-\ell_1+1}^{t-\ell_1-1} \alpha^i h(\alpha) \|C\|\|B\|.$$

Next we have

$$\left(\max_{\alpha(A)} \sum_{i=L-\ell_1+1}^{t-\ell_1-1} \alpha^i h(\alpha)\right) \leq h(\alpha)\alpha^{L-\ell_1-1} \sum_{i=0}^{T-L} \alpha^i$$

$$= h(\alpha)\alpha^{L-\ell_1-1} \frac{1 - \alpha^{T-L+1}}{1 - \alpha}$$

$$\leq T^{-1/4},$$

where the last inequality holds by assumption. Therefore Eq. 10 is at most

$$\|\sum_{i=1}^{T} M_i^{\text{true}}(u_{t-1:0} - u_{t-1:t-L}) v_i\| \leq \|C\|\|B\| T^{-1/4}.$$

Then we have

$$\|y_t - p_t(y_{-1:1}) - \sum_{i=1}^{k} M_i^{\text{true}} u_{t-1:t-L} v_i\|^2 \leq \frac{\|C\|^2\|B\|^2}{T^{1/2}} + 2\frac{\|C\|^2\|B\|^2}{T^{3/4}} + \frac{\|C\|^2\|B\|^2}{T^2} \leq 2\frac{\|C\|^2\|B\|^2}{T^{1/2}}.$$

Finally we conclude

$$\ell_t(M^{\text{true}} + E_M, L) \leq 2\frac{\|C\|^2\|B\|^2}{T^{1/2}} + 2\left(2\frac{\|C\|^2\|B\|^2}{T^{1/2}}\right)^{1/2}(k\delta\log^p(T)) + (k\delta\log^p(T))^2$$

$$\leq 4\frac{\|C\|^2\|B\|^2}{T^{1/2}},$$

where the last inequality holds since we assumed

$$\delta \leq \frac{1}{4m}\frac{\|C\|^2\|B\|^2}{T^{1/4}\log^p(T)}.$$

$\square$

## B  LENGTH GENERALIZATION FOR VANILLA SPECTRAL FILTERING

The proof of Theorem 5 ultimately comes from Theorem 7 and its proof in Appendix A. Theorem 7 abstracts the necessary assumptions needed to obtain a length generalization guarantee. In Lemma 12 we prove that Algorithm 1 satisfies these assumptions.

*Proof of Theorem 5.* By Lemma 12 and the assumptions made in the statement of Theorem 5, we may apply Theorem 7 to Algorithm 1 to get that

$$\sum_{t=1}^T \ell_t(M^t, L) - \min_{M^* \in \mathcal{K}_{\|C\|\|B\|}} \sum_{t=1}^T \ell_t(M^*, T) \leq \left(12k^{3/2}\|C\|^2\|B\|^2\log(T) + 8\|C\|^2\|B\|^2\right)\sqrt{T}.$$

$\square$

**Lemma 12** (Length Generalization for Vanilla Spectral Filtering). *Recall that in Algorithm 1 we define*

$$\mu_\alpha \overset{\text{def}}{=} (\alpha - 1)\begin{bmatrix} 1 & \alpha & \dots & \alpha^{T-1} \end{bmatrix}^\top \in \mathbb{R}^{T-1}$$

*and $H_{T-1} = \int_{\alpha \in [0,1]} \mu_\alpha \mu_\alpha^\top d\alpha$ and we let $\phi_1, \dots, \phi_{T-1}$ be the orthonormal eigenvectors of $H_{T-1}$ with eigenvalues $\sigma_1, \dots, \sigma_{T-1}$. Algorithm 1 is equivalent to Algorithm 3 with the following:*

*(a) $p_t(y_{t-1:1}) = y_{t-1}$*

*(b) $v_1 = e_1$*

*(c) $v_i = (0, \sigma_{i-1}^{1/4}\phi_{i-1})$ for $i = 2, \dots, T$*

*Define $M^{true}$ as follows:*

$$M_1^{true} \overset{\text{def}}{=} CB,$$

*and for $i \geq 2$*

$$M_i^{true} \overset{\text{def}}{=} \sum_{n=1}^{d_A} \sigma_{i-1}^{-1/4}\phi_{i-i}^\top \mu_{\alpha_n}(C_n B_n^\top).$$

*Then the following properties hold*

*1. For $h(A) = A - I$ and $\ell_1 = 1$*

$$y_t - p_t(y_{t-1:1}) = \sum_{i=1}^{\ell_1} M_i^{true} u_{t-i} + \sum_{i=1}^{t-\ell_1} CA^i h(A)Bu_{t-\ell_1-i}.$$

*2. $y_t - p_t(y_{t-1:1}) = \sum_{i=1}^T M_i^{true} u_{t-1:1} v_i$.*

3. For $k = \Omega(\log(T d_A \|C\|\|B\|/\epsilon))$,

$$\| \sum_{i=k+1}^{T} M_i^{true} u_{t-1:1} v_i \| \ \leq \ \epsilon/T.$$

4. For any $i \in [T]$

$$\|M_i^{true}\| \ \leq \ \|C\|\|B\|.$$

5. For any $i \in [T]$, $\|v_i\|_1 \ \leq \ \log(T)$ and $\{v_i\}_{i \in [T]}$ are orthonormal.

6. Finally if the spectrum of $A$ lies in the interval

$$\left[0, 1 - \frac{\log(T)}{2(L-2)}\right] \cup \left[1 - \frac{1}{2T^{5/4}}, 1\right],$$

then

$$\max_{\alpha(A)} \{|h(\alpha)\alpha^{L-\ell_1-1}(1-\alpha^{T-L+1})(1-\alpha)^{-1}|\} \ \leq \ \frac{1}{T^{1/4}}.$$

*Proof.* Points $(a) - (c)$ are evident by definition of Algorithm 1. Now suppose $y_t$ evolves as an LDS. By definition, there exist matrices $(A, B, C, D)$ such that

$$y_t = \sum_{i=1}^{t} CA^{i-1}Bu_{t-i},$$

where we assume $D = I$ and $A$ is diagonal without loss of generality. Let $\alpha_1, \dots, \alpha_{d_A}$ denote the eigenvalues of $A$. and let $u_{t:0}$ be the $d_{\text{in}} \times T$ (padded) matrix $u_{t:0} = [u_t \quad u_{t-1} \quad \dots \quad u_0 \quad 0]$. Then we have

$$y_t - y_{t-1} = \sum_{i=1}^{t} CA^{i-1}Bu_{t-i} - \sum_{i=1}^{t-1} CA^{i-1}Bu_{t-1-i}$$

$$= CBu_{t-1} + \sum_{i=1}^{t-1} C\left(A^i - A^{i-1}\right) Bu_{t-1-i}.$$

We pause here to note this proves $(1)$. We continue rearranging the equation to finish the derivation of $(2)$.

$$y_t - y_{t-1} = CBu_{t-1} + \sum_{i=1}^{t-1} C\left(A^i - A^{i-1}\right) Bu_{t-1-i}$$

$$= CBu_{t-1} + \sum_{n=1}^{d_A} Ce_n e_n^\top B \sum_{i=1}^{t-1} \left(\alpha_n^i - \alpha_n^{i-1}\right) u_{t-1-i}$$

$$= CBu_{t-1} + \sum_{n=1}^{d_A} (C_n B_n^\top) u_{(t-2):0} \mu_{\alpha_j}.$$

Observe that

$$\sum_{i=1}^{T-1} \phi_i \phi_i^\top = I.$$

Using this we have,

$$y_t - y_{t-1} = CBu_{t-1} + \sum_{n=1}^{d_A} (C_n B_n^\top) u_{(t-2):0} \mu_{\alpha_n}$$

$$= CBu_{t-1} + \sum_{n=1}^{d_A} (C_n B_n^\top) u_{(t-2):0} \left(\sum_{i=1}^{T} \phi_i \phi_i^\top\right) \mu_{\alpha_n}$$

$$= CBu_{t-1} + \sum_{i=1}^{T} \sum_{n=1}^{d_A} \phi_i^\top \mu_{\alpha_n} (C_n B_n^\top) u_{(t-2):0} \phi_i.$$

Recalling the definition of $M^{\text{true}}$ and $v_i = \sigma_{i-1}^{1/4}\phi_{i-1}$ we therefore have established (2):

$$y_t - y_{t-1} = M_1^{\text{true}} u_{(t-1):0} e_1 + \sum_{i=2}^{T-1} M_i^{\text{true}} u_{(t-1):0} v_i.$$

Next we aim to prove (3). We consider

$$\| \sum_{i=k+1}^{T} M_i^{\text{true}} u_{(t-2):0} v_i \|.$$

By Lemma 13.4 in Hazan & Singh (2022) there is some universal constant $c'$ such that,

$$\max_{\alpha \in [0,1]} |\phi_i^\top \mu_\alpha| \leq c' T^2 \exp(-i/\log(T)).$$

So,

$$\|M_i^{\text{true}} u_{(t-2):0} v_i\| = \| \sum_{n=1}^{d_A} \sigma_{i-1}^{-1/4} \phi_{i-i}^\top \mu_{\alpha_n} (C_n B_n^\top) u_{(t-2):0} \left( \sigma_{i-1}^{1/4} \phi_{i-1} \right) \|$$

$$= \| \sum_{n=1}^{d_A} \phi_{i-i}^\top \mu_{\alpha_n} (C_n B_n^\top) u_{(t-2):0} \phi_{i-1} \|$$

$$\leq d_A (c' T^2 \exp(-(i-1)/\log(T))) \|C_n B_n^\top\| \|\phi_{i-1}\|_1$$

$$\leq c' d_A T^{3/2} \exp(-(i-1)/\log(T))) \|C\| \|B\|.$$

Therefore,

$$\| \sum_{i=k+1}^{T} M_i^{\text{true}} u_{(t-1):0} \phi_i \| \leq c' d_A T^{5/2} \exp(-k/\log(T))) \|C\| \|B\|$$

Therefore as long as

$$k \geq \log(T) \log \left( \frac{T^{5/2} c' d_A \|C\| \|B\|}{\epsilon} \right),$$

then

$$\| \sum_{i=k+1}^{T} M_i^{\text{true}} u_{(t-1):0} \phi_i \| \leq \frac{\epsilon}{T}.$$

Next we note that the proof of (4) that $\|M_i^{\text{true}}\| \leq \|C\| \|B\|$ is proven in Lemma D.1 of Hazan et al. (2017b). Similarly, the proof of (5) that $\|v_i\|_1 \leq \log(T)$ is proven by Lemma 13 from Hazan et al. (2017b). Finally we prove (6). Since $h(\alpha) = \alpha - 1$ and $\ell_1 = 1$, we have

$$\max_{\alpha(A)} \left\{ |h(\alpha)\alpha^{L-\ell_1-1}(1 - \alpha^{T-L+1})(1 - \alpha)^{-1}| \right\} = \max_{\alpha(A)} \alpha^{L-2}(1 - \alpha^{T-L+1}). \tag{11}$$

To bound Eq. 11, consider the case where $\alpha$ is bounded away from 1. Suppose $\alpha = 1 - \delta$, then

$$(1-\delta)^{L-2} \leq \frac{1}{T^p} \iff \log \left( \frac{1}{1-\delta} \right) \geq \frac{p \log(T)}{L-2}.$$

Observe that for $\delta \in [0,1]$, $\log(1/(1-\delta)) \geq \delta/2$. Therefore, if

$$\delta \geq \frac{2p \log(T)}{L-2},$$

we are guaranteed that $\alpha^{L-2} \leq 1/T^p$. Next consider when $\alpha$ is very close to 1; suppose $\alpha \geq 1 - \frac{1}{T^p T}$ for $p < 1/2$. Then using that $(1-x)^q \geq 1 - 2qx$ for $x \in [0,1]$ we have

$$\alpha^{T-L+1} \geq \left( 1 - \frac{1}{T^p T} \right)^{T-L+1} \geq 1 - 2 \frac{T-L+1}{T^p T} \implies 1 - \alpha^{T-L+1} \leq 2 \frac{T-L+1}{T^p T} \leq \frac{2}{T^p}.$$

Plugging in $p = 1/4$ we conclude that

$$\alpha^{L-2}(1 - \alpha^{T-L+1}) \leq T^{-1/4} \qquad \text{for any } \alpha \in \left[0, 1 - \frac{\log(T)}{2(L-2)}\right] \cup \left[1 - \frac{1}{2T^{5/4}}, 1\right].$$

$\square$

The following lemma comes from Hazan et al. (2017b).

**Lemma 13** (Hazan, Singh, Zhang). *Let $(\sigma_j, \phi_j)$ be the $j$-th largest eigenvalue-eigenvector pair of the $T \times T$ Hankel matrix. Then,*

$$\|\phi_j\|_1 \leq O\left(\frac{\log(T)}{\sigma_j^{1/4}}\right).$$

## C LENGTH GENERALIZATION FOR SPECTRAL FILTERING USING TWO AUTOREGRESSIVE COMPONENTS

The proof of Theorem 6 ultimately comes from Theorem 7 and its proof in Appendix A. Theorem 7 abstracts the necessary assumptions needed to obtain a length generalization guarantee. In Lemma 14 we prove that Algorithm 2 satisfies these assumptions.

*Proof of Theorem 6.* By Lemma 14 and the assumptions made in the statement of Theorem 6, we may apply Theorem 7 to Algorithm 2 to get that

$$\sum_{t=1}^{T} \ell_t(M^t, L) - \min_{M^* \in \mathcal{K}_{\|C\|\|B\|}} \sum_{t=1}^{T} \ell_t(M^*, T) \leq \left(12k^{3/2}\|C\|^2\|B\|^2 \log^2(T) + 8\|C\|^2\|B\|^2\right) \sqrt{T}.$$

$\square$

**Lemma 14** (Length Generalization Using Two Autoregressive Components). *Recall that in Algorithm 2 we define*

$$\tilde{\mu}_{\alpha,T} \stackrel{def}{=} (\alpha - 1)^2 \begin{bmatrix} 1 & \alpha & \dots & \alpha^T \end{bmatrix}^\top \in \mathbb{R}^T$$

*and and $N_T = \int_{\alpha \in [0,1]} \tilde{\mu}_{\alpha,T} \tilde{\mu}_{\alpha,T}^\top d\alpha$ and we let $\tilde{\phi}_1, \dots, \tilde{\phi}_{T-2}$ be the orthonormal eigenvectors of $N_{T-2}$ with eigenvalues $\tilde{\sigma}_1, \dots, \tilde{\sigma}_{T-2}$. Algorithm 2 is equivalent to Algorithm 3 with the following:*

*(a) $p_t(y_{t-1:1}) = 2y_{t-1} - y_{t-2}$*

*(b) $v_1 = e_1$, $v_2 = e_2$ and for $i \geq 3$, $v_i = (0, 0, \sigma_{i-2}^{1/4} \tilde{\phi}_{i-2})$*

*Define $M^{true}$ as follows:*

$$M_1^{true} \stackrel{def}{=} CB,$$

$$M_2^{true} \stackrel{def}{=} C(A - 2I)B,$$

*and for $i \geq 3$,*

$$M_i^{true} \stackrel{def}{=} \sum_{n=1}^{d_A} \left(\sigma_i^{-1/4} \tilde{\phi}_i^\top \tilde{\mu}_{\alpha_n}\right)(C_n B_n^\top).$$

*Then the following properties hold*

*1. For $h(A) = (A - I)^2$ and $\ell_1 = 2$*

$$y_t - p_t(y_{t-1:1}) = \sum_{i=1}^{\ell_1} M_i^{true} u_{t-i} + \sum_{i=1}^{t-\ell_1} CA^i h(A) Bu_{t-\ell_1-i}.$$

2. $y_t - p_t(y_{t-1:1}) = \sum_{i=1}^{T} M_i^{true} u_{t-1:1} v_i.$

3. *For $k = \Omega(\log(T d_A \|C\| \|B\|/\epsilon))$,*

$$\| \sum_{i=k+1}^{T} M_i^{true} u_{t-1:1} v_i \| \leq \epsilon/T.$$

4. *For any $i \in [T]$*

$$\|M_i^{true}\| \leq \|C\| \|B\|.$$

5. *For any $i \in [T]$, $\|v_i\|_1 \leq \log(T)$ and $\{v_i\}_{i \in [T]}$ are orthonormal.*

6. *Finally if the spectrum of $A$ lies in the interval*

$$\left[0, 1 - \frac{\log(T)}{2(L-2)}\right] \cup \left[1 - \frac{1}{2T^{1/4}}, 1\right],$$

*then*

$$\max_{\alpha(A)} \left\{ |h(\alpha)\alpha^{L-\ell_1-1}(1 - \alpha^{T-L+1})(1-\alpha)^{-1}| \right\} \leq \frac{1}{T^{1/4}}.$$

*Proof.* Suppose $y_t$ evolves as an LDS. By definition, there exist matrices $(A, B, C, D)$ such that

$$y_t = \sum_{i=1}^{t} CA^{i-1} B u_{t-i},$$

where we assume $D = I$ and $A$ is diagonal without loss of generality. Let $\alpha_1, \ldots, \alpha_{d_A}$ denote the eigenvalues of $A$. and let $u_{t:0}$ be the $d_{in} \times T$ (padded) matrix $u_{t:0} = [u_t \quad u_{t-1} \quad \ldots \quad u_0 \quad 0]$. Then we have (1):

$$y_t - 2y_{t-1} + y_{t-2} = CBu_{t-1} + C(A - 2I)Bu_{t-2} + \sum_{i=0}^{t-3} CA^i(A^2 - 2A + I)Bu_{t-3-i}.$$

Let $\alpha_1, \ldots, \alpha_{d_A}$ denote the eigenvalues of $A$. We observe the following equality:

$$\sum_{i=0}^{t-3} CA^i(A^2 - 2A + I)Bu_{t-3-i} = \sum_{i=0}^{t-3} C \sum_{n=1}^{d_A} \alpha_n^i (\alpha_n - 1)^2 e_n e_n^\top Bu_{t-3-i}$$

$$= \sum_{n=1}^{d_A} \left(Ce_n e_n^\top B\right) \sum_{i=0}^{t-3} \alpha_n^i (\alpha_n - 1)^2 u_{t-3-i}$$

$$= \sum_{n=1}^{d_A} \left(C_n B_n^\top\right) u_{(t-3):0} \tilde{\mu}_{\alpha_n}.$$

Observe that

$$\sum_{i=1}^{T-2} \tilde{\phi}_i \tilde{\phi}_i^\top = I.$$

Using this we have,

$$\sum_{i=0}^{t-3} CA^i(A^2 - 2A + I)Bu_{t-3-i} = \sum_{n=1}^{d_A} \left(C_n B_n^\top\right) u_{(t-3):0} \tilde{\mu}_{\alpha_n}$$

$$= \sum_{n=1}^{d_A} \left(C_n B_n^\top\right) u_{(t-3):0} \left(\sum_{i=1}^{T-2} \tilde{\phi}_i \tilde{\phi}_i^\top\right) \tilde{\mu}_{\alpha_n}$$

$$= \sum_{i=1}^{T-2} \left(\sum_{n=1}^{d_A} \tilde{\phi}_i^\top \tilde{\mu}_{\alpha_n} \left(C_n B_n^\top\right)\right) u_{(t-3):0} \tilde{\phi}_i$$

$$= \sum_{\ell=3}^{T} M_\ell^{true} u_{(t-1):0} v_\ell.$$

Therefore we have established (2). Next we aim to prove (3). We consider

$$\|\sum_{i=k+1}^{T} M_i^{\text{true}} u_{(t-1):0} v_i\|.$$

Combining Lemma 15 and Lemma 16 gives us that there is some constant $c'$ such that,

$$\max_{\alpha \in [0,1]} |\tilde{\phi}_i^\top \tilde{\mu}_\alpha| \leq c' \exp(-i/4 \log(T)).$$

So,

$$\|M_i^{\text{true}} u_{(t-1):0} v_i\| = \|\sum_{n=1}^{d_A} \sigma_{i-1}^{-1/4} \tilde{\phi}_{i-i}^\top \tilde{\mu}_{\alpha_n} (C_n B_n^\top) u_{(t-1):0} \left(\sigma_{i-1}^{1/4} \tilde{\phi}_{i-1}\right) \|$$

$$= \|\sum_{n=1}^{d_A} \tilde{\phi}_{i-i}^\top \tilde{\mu}_{\alpha_n} (C_n B_n^\top) u_{(t-2):0} \tilde{\phi}_{i-1}\|$$

$$\leq d_A \exp(-(i-1)/4 \log(T)) \|C_n B_n^\top\| \|\phi_{i-1}\|_1$$

$$\leq c' d_A \sqrt{T} \exp(-(i-1)/4 \log(T)) \|C\| \|B\|.$$

Therefore,

$$\|\sum_{i=k+1}^{T} M_i^{\text{true}} u_{(t-1):0} v_i\| \leq c' d_A T^{3/2} \exp(-i/4 \log(T)) \|C\| \|B\|.$$

Therefore as long as

$$k \geq 4 \log(T) \log\left(\frac{T^{3/2} c' d_A \|C\| \|B\|}{\epsilon}\right),$$

then

$$\|\sum_{i=k+1}^{T} M_i^{\text{true}} u_{(t-1):0} v_i\| \leq \frac{\epsilon}{T}.$$

To prove (4) we note that the statement is obvious for $i \leq 2$. For $i \geq 3$ the proof from Lemma D.1 of Hazan et al. (2017b) directly applies due to Lemma 15. Next, Lemma 17 proves (5). Finally we prove (6). Next, Lemma 17 proves (5). Finally we prove (6). Since we have $h(\alpha) = (\alpha - 1)^2$ and $\ell = 2$,

$$\max_{\alpha(A)} \left\{ |h(\alpha) \alpha^{L-3} (1 - \alpha^{T-L+1})(1-\alpha)^{-1}| \right\} = \max_{\alpha(A)} \left\{ (1-\alpha) \alpha^{L-3} (1 - \alpha^{T-L+1}) \right\}. \qquad (12)$$

To bound Eq. 12, consider the case where $\alpha$ is bounded away from 1. Suppose $\alpha = 1 - \delta$, then

$$(1-\delta)^{L-3} \leq \frac{1}{T^p} \iff \log\left(\frac{1}{1-\delta}\right) \geq \frac{p \log(T)}{L-3}.$$

Observe that for $\delta \in [0,1]$, $\log(1/(1-\delta)) \geq \delta/2$. Therefore, if

$$\delta \geq \frac{2p \log(T)}{L-3},$$

we are guaranteed that $\alpha^{L-3} \leq 1/T^p$. Next consider when $\alpha$ is very close to 1. To ensure that Eq. **??** is bounded by $1/T^p$ we only require

$$\alpha \geq 1 - \frac{1}{T^p}.$$

Plugging in $p = 1/4$, we conclude that Eq. **??** is bounded by $T^{-1/4}$ if

$$\alpha_n \in \left[0, 1 - \frac{\log(T)}{2(L-3)}\right] \cup \left[1 - \frac{1}{T^{1/4}}, 1\right] \text{ for all } n \in [d_A].$$

□

## C.1 Properties of the Hankel Matrix for Two Autoregressive Terms

In Algorithm 2 we define

$$\tilde{\mu}_\alpha \stackrel{\text{def}}{=} (\alpha - 1)^2 \begin{bmatrix} 1 & \alpha & \dots & \alpha^T \end{bmatrix}^\top \in \mathbb{R}^T$$

and

$$N_T = \int_{\alpha \in [0,1]} \tilde{\mu}_\alpha \tilde{\mu}_\alpha^\top d\alpha.$$

In what follows we present and prove several lemmas needed for the proof of Theorem 6.

**Lemma 15** (Properties of $N_T$). *For any $\alpha \in [0,1]$ and $1 \le i \le T$,*

$$\max_{\alpha \in [0,1]} |\phi_i^\top \tilde{\mu}_\alpha| \le 6^{1/4} \sigma_i^{1/4}.$$

*Proof.* We have

$$\int_{\alpha \in [0,1]} \left( \phi_i^\top \tilde{\mu}_\alpha \right)^2 d\alpha = \phi_i^\top \left( \int_{\alpha \in [0,1]} \tilde{\mu}_\alpha \tilde{\mu}_\alpha^\top d\alpha \right) \phi_i$$

$$= \phi_i^\top N_T \phi_i = \sigma_i.$$

Next we observe that for $f_w(\alpha) \stackrel{\text{def}}{=} \left( w^\top \tilde{\mu}_\alpha \right)^2$, where $w$ is any unit-norm vector, we have that $f_w$ is 6-Lipschitz on $[0,1]$. Indeed,

$$f_w'(\alpha) = \frac{d}{d\alpha} (\alpha - 1)^4 \left( \sum_{i=1}^T w_i \alpha^{i-1} \right)^2$$

$$= 2(\alpha - 1)^4 \left( \sum_{i=1}^T w_i \alpha^{i-1} \right) \left( \sum_{i=2}^T (i-1) w_i \alpha^{i-2} \right) + 4 \left( \sum_{i=1}^T w_i \alpha^{i-1} \right)^2 (\alpha - 1)^3$$

$$\le 2(\alpha - 1)^4 \left( \frac{1 - \alpha^T}{1 - \alpha} \right) \left( \sum_{i=1}^{T-1} i \alpha^{i-1} \right) + 4 \left( \frac{1 - \alpha^T}{1 - \alpha} \right)^2 (\alpha - 1)^3$$

$$= 2(\alpha - 1)^4 \left( \frac{1 - \alpha^T}{1 - \alpha} \right) \left( \frac{1 - T\alpha^{T-1} + (T-1)\alpha^T}{(1 - \alpha)^2} \right) + 4 \left( \frac{1 - \alpha^T}{1 - \alpha} \right)^2 (\alpha - 1)^3$$

$$= 2 \left( 1 - \alpha^T \right) \left( 1 - T\alpha^{T-1} + (T-1)\alpha^T \right) + 4 \left( 1 - \alpha^T \right)^2 (\alpha - 1)$$

$$\le 2 + 4 = 6.$$

Consider any non-negative L-Lipschitz function $f$ that reaches some maximum value $g_{\max}$ over $[0,1]$. The function $f$ which satisfies $L$-Lipschitzness, attains $g_{\max}(f)$ and also has minimum possible area $A(f) \stackrel{\text{def}}{=} \int_{\alpha \in [0,1]} f(\alpha) d\alpha$ is

$$f^*(\alpha) = \begin{cases} L\alpha, & \text{for } \alpha \in [0, \alpha^*] \\ \max \{ g_{\max} - L(\alpha - \alpha^*), 0 \}, & \text{for } \alpha \in [\alpha^*, 1] \end{cases}$$

$$= \begin{cases} L\alpha, & \text{for } \alpha \in [0, \alpha^*] \\ g_{\max} - L(\alpha - \alpha^*), & \text{for } \alpha \in [\alpha^*, \alpha^* + \frac{g_{\max}}{L}] \\ 0, & \text{for } \alpha \in [\alpha^* + \frac{g_{\max}}{L}, 1] \end{cases}.$$

Indeed, any oscillation away from this piecewise linear function would either increase the total area or violate the Lipschitz constraint. For this to be a valid construction we must have $L\alpha^* = g_{\max}$ and therefore the minimum corresponding area is

$$A(f^*) = \int_{\alpha \in [0,1]} f^*(\alpha) d\alpha = \frac{1}{2} (\alpha^*)(L\alpha^*) + \frac{1}{2} (g_{\max}/L) g_{\max} = \frac{g_{\max}^2}{L}.$$

And therefore for any function $f$ we have $g_{\max}(f) \leq \sqrt{LA(f)}$. Using this for $f_{\phi_i}(\alpha)$ we have

$$\max_{\alpha \in [0,1]} f_{\phi_i}(\alpha) = \max_{\alpha \in [0,1]} (\phi_i^\top \tilde{\mu}_\alpha)^2 \leq \sqrt{6 \int_{\alpha \in [0,1]} \left(\phi_i^\top \tilde{\mu}_\alpha\right)^2 d\alpha} = \sqrt{6\sigma_i}.$$

We conclude by noting

$$\max_{\alpha \in [0,1]} |\phi_i^\top \tilde{\mu}_\alpha| = \sqrt{\max_{\alpha \in [0,1]} (\phi_i^\top \tilde{\mu}_\alpha)^2} \leq 6^{1/4} \sigma_i^{1/4}.$$

$\square$

**Lemma 16** (Adapted from Lemma $E.2$ from Hazan et al. (2017b)). *Let $\sigma_j$ be the $j$-th top singular value of $N_T$. Then for all $T \geq 10$ we have*

$$\sigma_j \leq \min\left(\frac{3}{2}, K \cdot c^{-j/\log(T)}\right),$$

*where $c = e^{\pi^2/4} \approx 11.79$ and $K \leq 10^6$ is an absolute constant.*

*Proof.* The proof provided in Hazan et al. (2017b) applies directly to $N_T$ with only one necessary modification to bound the trace. Observe that we have

$$(N_T)_{ij} = \int_{\alpha \in [0,1]} (\alpha - 1)^4 \alpha^{i+j-2} d\alpha$$

$$= \int_{\alpha \in [0,1]} \alpha^{i+j} - 2\alpha^{i+j-1} + \alpha^{i+j-2} d\alpha$$

$$= \frac{24}{(i+j-1)(i+j)(i+j+1)(i+j+2)(i+j+3)}.$$

Therefore,

$$\sigma_j \leq \operatorname{tr}(N_T) = \sum_{i=1}^{T} \frac{24}{(2i-1)(2i)(2i+1)(2i+2)(2i+3)} \leq \sum_{i=1}^{T} \frac{24}{(2i)^5} = \frac{3}{4} \sum_{i=1}^{T} \frac{1}{i^5} < \frac{3}{2}.$$

The remainder of the proof is an exact copy of the proof of Lemma $E.2$ with $3/4$ replaced by $3/2$. $\square$

**Lemma 17** (Controlling the $\ell_1$ norm of the filters). *Let $(\sigma_j, \phi_j)$ be the $j$-th largest eigenvalue-eigenvector pair of $N_T$. Then for $T \geq 4$,*

$$\|\phi_j\|_1 \leq O\left(\frac{\log T}{\sigma_j^{1/4}}\right).$$

*Proof.* This proof is a copy from the proof of Lemma E.5 in Hazan et al. (2017b) with only one noted modification. We note that $E$ as defined in their proof is entrywise bounded (for $T \geq 4$) by $24/T^5 \leq 2/T^3$ (which is the stated bound they use for their matrix of interest). We also must show the base case is true for $T_0 = 4$ instead of $T_0 = 2$. We have

$$\|N_4^{1/4}\|_{2\to 1} = \sup_{x:\|x\|_2 \leq 1} \|N_4^{1/4} x\|_1 \leq \sum_{i,j=1}^{4} |\left(N_4^{1/4}\right)_{ij}| < 2.$$

We note that a tighter result is actually true for $N_T$ in that $\|\phi_j\|_1 \leq O\left(\frac{\log T}{\sigma_j^{1/8}}\right)$. However, we omit this statement and proof because we don't leverage it for a tighter result overall.

$\square$

In Algorithm 2 we define

$$\tilde{\mu}_\alpha \overset{\text{def}}{=} (\alpha - 1)^2 \begin{bmatrix} 1 & \alpha & \dots & \alpha^T \end{bmatrix}^\top \in \mathbb{R}^T$$

and

$$N_T = \int_{\alpha \in [0,1]} \tilde{\mu}_\alpha \tilde{\mu}_\alpha^\top d\alpha.$$

We have

$$(N_T)_{ij} = \int_{\alpha \in [0,1]} (\alpha - 1)^4 \alpha^{i+j-2} d\alpha$$

$$= \int_{\alpha \in [0,1]} \alpha^{i+j} - 2\alpha^{i+j-1} + \alpha^{i+j-2} d\alpha$$

$$= \frac{24}{(i+j-1)(i+j)(i+j+1)(i+j+2)(i+j+3)}.$$