# OpenReview forum: "Provable Length Generalization in Sequence Prediction via Spectral Filtering"
_ICLR.cc/2025/Conference — Submitted to ICLR 2025_

### Official Review · Reviewer_Q1Zn · 2024-10-25

**Soundness:** 3
**Presentation:** 3
**Contribution:** 2
**Rating:** 8
**Confidence:** 3

**Summary:**

This paper studies the problem of predicting the outputs of a linear dynamic system using only a truncated history. The paper shows that the regret of a predictor with history length $T^q$ is upper bounded by $\tilde{O}(\sqrt{T})$ compared to the full-length history (i.e., length $T$) experts, provided the data is noiseless and follows some underlying linear dynamic system. This is achieved at the cost of restricting the eigenvalue (depending on $q$) of the leading matrix $A$ of the linear dynamic system.

**Strengths:**

In my opinion, the primary contribution of the paper is conceptual; it shows that meaningful regret bounds can still be achieved even when the predictor is not exposed to the full history, at least in the context of linear dynamic systems. This may inspire future work to extend this idea to other settings. As the authors point out, this truncated scenario is also inspired by the limited token-length prediction in LLMs (although this is not directly related to the current paper).

**Weaknesses:**

The proof idea appears to me to be quite simple, essentially involving the application of an online convex optimization predictor on the *truncated* expert. The regret bound follows from the fact that the performance of the optimal truncated expert is close to that of the optimal full-history predictor (using the assumption that the eigenvalues of $A$ are bounded). The final regret bound is essentially a restatement of Hazan et al. (2017b).

As I am not an expert in linear dynamic system theory, I cannot comment on the significance of relating truncated experts to full-history experts in that field. In particular, I am unsure how restrictive it is to constrain the eigenvalues of $A$.

From an online learning perspective, the technical contribution is minimal (both algorithmically and analytically), as the analysis closely follows Hazan et al. (2017b).

**Questions:**

1. It seems your result holds only in the noiseless setting. What happens if noise is introduced? Does the regret bound of Hazan et al. (2017b) hold in this case as well?

2. What happens if we allow the predictor to compute a "sketch" of the history, rather than a hard truncation?

---

> ### Author Response · Authors · 2024-11-23
> **Responses to Weaknesses and Questions**
>
> ## Responses to Weaknesses
> 1. Please check the main comment to all reviewers where we address this.
> 2. Assumptions on the eigenvalues of A are important. There are several cases: 1) eigenvalues of A are allowed to be both real and complex, 2) eigenvalues of A are allowed to be nonnegative, 3) eigenvalues of A are allowed to be greater than 1. For (1) there aren’t any special methods that provably work to our knowledge. You can apply classical methods of prediction like regression, etc. However, to obtain meaningful results you must assume marginal stability (i.e. the magnitude of the eigenvalues is strictly bounded by 1). For (2) vanilla spectral filtering provably achieves O(sqrt(T)) regret and it is possible to adapt our result to extend to this case. For (3), there is nothing to do because the system is not stable and the values diverge as T gets large (moreover in the regime where the values are bounded given a horizon T, then rescaling them to be bounded by 1 can be done without loss of generality).
> 3. Please check the main comment to all reviewers where we address this.
>
> ## Responses to Questions
> 1. The regret bound in Hazan et al. (2017b) does not make assumptions on the data, however it only compares to the best LDS predictor (which is by virtue noiseless) and therefore it does not imply that spectral filtering predicts well in general on an LDS with noise. In general LDS literature, Kalman filtering is bayes-optimal with gaussian noise (however this is only for stable systems i.e. eigenvalues of A are bounded away from 1 and therefore the memory of the system is bounded).
> 2. This is a very cool idea and to our knowledge the empirical methods designed to handle length generalization for LLMs in practice don’t currently try to do this. In a more general sense, theoretical analysis of how best to use memory resources for sequence prediction is an interesting question.

---

> > ### Comment · Reviewer_Q1Zn · 2024-11-23
> >
> > I appreciate the authors for clarifying the necessity of constraining the eigenvalues of $A$ and for their effort in improving the presentation regarding the technical novelty. Since my concern has been generally resolved, I am increasing my score to 8. I believe this paper is a valuable contribution to the community.

---

### Official Review · Reviewer_f77n · 2024-11-03

**Soundness:** 2
**Presentation:** 2
**Contribution:** 2
**Rating:** 5
**Confidence:** 2

**Summary:**

The paper presents a novel approach to length generalization in sequence prediction using an Asymmetric-Regret metric. This metric measures prediction performance for models constrained to shorter context lengths, compared to a benchmark with access to the full sequence history. The authors apply this metric within spectral filtering for linear dynamical systems (LDS) and introduce a gradient-based learning algorithm that maintains sublinear Asymmetric-Regret bounds, enabling generalization over long sequences even with limited training contexts.
This work builds on foundational contributions by Hazan and colleagues, who developed algorithms for spectral filtering in LDS under online convex optimization frameworks. The authors extend Hazan’s ideas by focusing specifically on achieving Asymmetric-Regret bounds with shorter context lengths.

**Strengths:**

1.	The paper extends the Asymmetric-Regret framework by applying it to limited context settings in spectral filtering. This adaptation addresses the relatively unexplored problem of length generalization for sequence prediction with limited context.
2.	Algorithm 1 introduces a modification of spectral filtering using shorter context windows, providing a new approach to minimize regret relative to models using the full sequence.

**Weaknesses:**

1.	Strong Assumptions:The main result relies on restrictive assumptions about LDS structures and eigenvalue constraints (e.g., eigenvalues must fall within “good” ranges), which may limit the algorithm’s applicability in situations that do not strictly follow LDS dynamics or contain eigenvalues outside these ranges.

2.	Limited Empirical Validation: The empirical scope is narrow, with experiments focused primarily on synthetic LDS data and a single deep learning task. This raises concerns about the method’s practical utility and performance in diverse real-world settings.

3.	Conceptual Novelty and Integration of Existing Ideas: While the application of Asymmetric-Regret bounds to shorter contexts is novel, the overall contribution seems primarily integrative, synthesizing elements from Hazan’s work on spectral filtering, convex optimization, and regret minimization. The empirical and conceptual innovations feel incremental rather than groundbreaking, especially given the limited empirical exploration.

**Questions:**

1.	Generalizability: Could the authors discuss how their approach might extend to more general settings, especially those where data does not conform strictly to LDS assumptions?
2.	Handling “Bad” Eigenvalues: Theoretical and empirical results suggest that the approach struggles when eigenvalues fall within certain “bad” ranges. Could the authors elaborate on adjustments to handle such cases?
3.	Comparison with Full-Context Models: In practice, how closely do regret bounds from shorter context lengths compare to those obtained with full sequence contexts? Are the additional comparative metrics that are relevant here?
4.	Empirical Testing Beyond Synthetic Data: This is needed to provide insights into the approach’s practical feasibility and generalizability.
5.	Alignment with Hazan’s Work: How does the adaptation of the Asymmetric-Regret metric aligns with or diverges from Hazan’s original work? A detailed and systematic discussion is needed to understand the key contributions of this work.

---

> ### Author Response · Authors · 2024-11-23
> **Responses to Weaknesses & Questions**
>
> ## Responses to Weaknesses
> 1. This is no longer the case, please see the main note to all authors.
> 2. In this work, we use the experiments mainly to validate our theory (on LDS) and to highlight the factors at play (context length, memory of the system, etc.). As mentioned in the response to another reviewer, LDS-based models are growing in popularity and perform competitively in practical and diverse real-world applications. It is not the goal of our experiments to convince readers that LDS methods have broad utility and behave well as layers in deep models, but rather to exemplify that a particular LDS method (spectral filtering) has an intrinsic efficient length generalization ability. The simple deep learning task is included as a proof-of-concept to suggest that this natural length generalization appears in such a setting. We consider it an important future work to perform a thorough empirical investigation into the right way to take advantage of this and integrate on real-world tasks, but this is a modeling question and is complementary to our results. In short, the practical utility of LDS-based models is already known – we prove length generalization for a particular LDS prediction algorithm and experimentally hint that its benefit carries over to LDS-based deep models, and leave the integration into real-world solutions for future work.
> 3. Please see our main note to reviewers regarding the nontriviality of the theoretical results.
>
> ## Responses to Questions
> 1. For more general sequence prediction tasks, such as nonlinear dynamics or language modeling, it has been empirically shown that one can design spectral filtering-based models that perform very competitively (see Flash STU paper https://arxiv.org/abs/2409.10489). Our results here demonstrate an inherent length generalization property in the way that spectral filtering predicts (in the sense that the optimal predictor on a short context is still optimal given the full one, which is not at all true for general sequence prediction algorithms). While we are unable to prove that these spectral filtering-based deep learning models achieve the same asymmetric-regret bounds on language modeling, for example, it is natural to expect that using a layer that can generalize results in an overall model with improved length generalization capability. We leave it to future work to ask the serious deep learning modeling questions and investigate the length generalization effects in large models, and we included a synthetic deep learning task (induction heads) as a minor first step in this direction.
> 2. Indeed, we have designed a new algorithm which uses two autoregressive components that provably is robust to this bad range. Not only do we provide a theoretical guarantee of this, we show experimentally that this approach is able to perform well on those bad regions.
> 3. The current upper bound for spectral filtering using the full context  is O(sqrt(T)). This is why we say we achieve “length-generalization”-- we mean that we achieve the same regret bound despite using much smaller context length.
> 4. We absolutely agree that a full-scale investigation into these length generalization behaviors on real-world tasks (such as language modeling or dynamics prediction) is crucial. Answering this fully is out of the scope of our current theoretical work: instead, we tried to get a first-pass indication of these behaviors on induction heads, which is synthetic and approachable yet considerably outside the assumptions of our theory. However, as mentioned in a response to reviewer xrE3, the topic of evaluating length generalization empirically gets unclear quickly and varies with modeling decisions and choices of datasets/tasks. At the end of the day, what matters is the behavior at the largest scales when the model is pushed to the limit (such as in large language models), on modes of length generalization that matter in practice (such as summarization of very long documents during deployment). There are prior works (e.g. the Flash-STU paper) that indicate STU models are competitive at such scales, and our current work is complementary in that it proves a precise theoretical reason to expect these already-successful models to length-generalize. Properly verifying this at the useful scales is a third contribution to the story, which we leave to future research – without a large experimental setup, it would be difficult to test this conjecture in any meaningful way, and so we treat the synthetic experiment strictly as a proof-of-concept.
> 5. Hazan’s work just considers the notion of regret that is standard in the literature. We have added a bit more on this after Definition 3 of asymmetric regret in Section 2.1. Please let us know if you have questions about asymmetric regret or standard regret since these definitions are key to understanding the work.

---

### Official Review · Reviewer_xrE3 · 2024-11-03

**Soundness:** 2
**Presentation:** 3
**Contribution:** 2
**Rating:** 3
**Confidence:** 4

**Summary:**

This paper studies length generalization in linear dynamical systems from the perspective of spectral filtering.  Length generalization, the phenomenon by which models trained to conduct algorithmic tasks on short sequences can then accurately apply the same algorithmic principles to longer inputs, has seen much recent interest in language models.  The authors examine a similar problem in the context of noiseless LDSs, where they define length generalization though a new notion of regret they call Asymmetric Regret that allows the comparator policy to incorporate a longer history than the learner can.  The authors leverage recent results on spectral filtering to demonstrate that some marginally stable linear systems can be learned (in the sense of Asymmetric Regret) using a natural online empirical risk minimization algorithm defined by small Frobenius norm linear functions on a basis given by spectral filters.  The authors then demonstrate that this algorithm attains good Asymmetric Regret as long as the spectrum of the eigenvalues of the system matrix A satisfies a certain technical assumption.  Finally, the authors conclude with two experiments: first showing that their assumption on the spectrum of A appears to be necessary by considering synthetic data from randomly generated LDS's and second showing that a certain neural architecture motivated by spectral filtering appears to length generalize when considering induction heads.

**Strengths:**

This paper sets out to study length generalization and proposes a new definition that formalizes the concept, at least in the task of sequential prediction.  Furthermore, they demonstrate that in the context of noiseless LDS', a natural algorithm based on earlier work in linear control succeeds in length generalization according to their proposed definition.  The proofs are correct as written and clearly presented.

**Weaknesses:**

I think the work has several major weaknesses:

1. The authors claim their study is motivated by the challenge of length generalization in LLMs, but they did not spend sufficient time justifying the connection between this motivation and their actual results.  I will discuss the empirical aspects below, but on the level of theory, it is not completely clear to me to what extent the study of online prediction in LDS's provides intuition for LLMs.  I think one way of connecting these two disparate concerns would be to note that because LLMs are trained with next token prediction, one may think of generation as an online prediction problem and thus Asymmetric Regret is a natural target notion as the LM is trained to produce algorithms with good `symmetric regret' on algorithmic tasks of interest, but, in order to length-generalize in a meaningful sense, small asymmetric regret (with respect to the longer sequence) is necessary.  If this is the way the authors are thinking about it, it would be helpful if they were to spell this out.  Assuming that this is correct, I guess I am a little confused as to what this means for length generalization in the context of LLMS?  It is not at all obvious to me that the theoretical results necessarily suggest that the Spectral Transform Unit of Agarwal et al 2023 is helpful in length generalization (at least beyond noiseless linear dynamical systems).  What broader intuition are we supposed to glean from the main theorem?

2. I think that the strength of the technical contribution may not quite meet the high bar set by the standards of the conference.  While it is nice that the authors introduce this notion of Asymmetric Regret, it is not entirely clear to what extent it is meaningful beyond the setting of LDS'.  Furthermore, on the level of proof techniques, as said in the paper, Lemma 5 is essentially standard.  Lemma 6 follows from elementary computations on top of earlier work by Hazan et al 2017 and standard results in the field of online control.  Finally, the main theorem is immediate from applying these two lemmata.  Thus, I am a little unclear as to the extent of the technical contribution beyond the introduction of the Asymmetric Regret.  Furthermore, it might be helpful for the authors to discuss alternative notions of regret from the online learning literature, as I think Asymmetric Regret can be thought of as a special case of the $\Phi$-regret introduced in Stolz and Lugosi 2007 and studied in *Online Learning: Beyond Regret* by Rakhlin, Sridharan, and Tewari 2011, where the $\Phi$ functions map a function class allowed to see the entire history to one constrained to seeing the most recent $T^q$ inputs.

3. The main theorem rests on a fairly unmotivated assumption on the spectrum of a system matrix and there is no theory regarding whether or not this is necessary.

4.  While I am reviewing this paper primarily as a theoretical one, I do wish to mention that I am not entirely convinced by the experiments on the STU networks.  First, the results in Figure 5 appear extremely noisy.  Second, and more importantly, there is no comparison to comparable results with transformers.  My understanding of some recent empirical results in the length generalization literature, such as *What Algorithms can Transformers Learn? A Study in Length Generalization* by Zhou et al 2023, is that induction heads should be relatively easy for transformers to learn due to their ease of representation; why do we expect STUs to improve in this setting?

**Questions:**

1. Can the authors present some intuition as to the assumption on the spectrum that is required in the main theorem?  The precise union of intervals seems extremely unmotivated to me.  While I understand that it is qualitatively reasonable in the sense that as the amount of data becomes smaller, the assumption becomes stricter, I do not have a lot of intuition beyond that.

2. While I realize that the authors present some empirical evidence suggesting that the main theorem's spectral assumption is necessary, can they provide any theoretical lower bounds for their algorithm?

3. How can the accuracy in Figure 5 be greater than 1?

---

> ### Author Response · Authors · 2024-11-23
> **Responses to Weaknesses**
>
> ## Response to Weakness 1
> We first note that inspired by the above we have added some extra text in the introduction (highlighted in blue) to address the connection between LDSs and LLMs. We hesitate to write much more since it may make a future reader think the point of the paper is LLMs when it isn’t. Please let us know if it is helpful. We now directly answer the reviewer.
>
> First, in our related work we reference 11 papers that address the connection between the study of online prediction in LDSs and LLMs (see “Motivated by the high memory and computer requirements of transformers, state space models were revisited…"). However, we implicitly assumed the reader would know that SSMs and the work we cited are based on considering online prediction in LDSs and we now make this connection explicit to the reader who is less familiar with this area of work. These works indeed show that methods based on studying LDSs provide empirically strong approaches in LLM applications. Thank you for pointing this out.
>
> Second, your intuition regarding asymmetric regret and how it may imply the STU architecture is helpful in length generalization for LLMs is correct. However, this isn’t the main point of the paper and to make this point fully we are planning to do a fully experimental work. That said, we now make this point clearer in the introduction (see the blue text).
>
> Finally, in response to what broader intuition should be gleaned from the main theorem: the fact that there exists a computable algorithm (the algorithms we introduce in our paper) which can use only T^{1/4} context length to both learn and make its predictions and obtain small regret compared to the best predictor using the full history of size T is very surprising. This result implies that spectral filtering is powerful in its ability to learn the dynamics of a complicated underlying system with long memory since it naturally handles the issue of what aspects in a sequence should be memorized for the future and what aspects can be forgotten, whereas other existing methods are hand engineered depending on the specific task. Thank you for highlighting that it isn’t clear to the reader. We have added a bit of this intuition into the discussion.
>
> ## Response to Weakness 2
> The fact that it isn’t clear to what extent this result is meaningful beyond LDSs makes sense given your question regarding the use of studying LDSs and their connection to language models (see above the discussion to your first point), which we will make clear. Indeed, results concerning LDSs tend to be quite powerful and useful in practice. For the second part of this weakness, please see our main note to reviewers regarding the nontriviality of the theoretical results. Regarding \phi-regret, it was first introduced by Greenwald and Jafari in 2003, and the context in which it was studied (by the references you cite, as well as Hazan and Kale 2007) is convergence to various notions of equilibrium and relationship to computation of stationary points. We are happy to elaborate on the relationship between this notion and asymmetric regret, which deals primarily with information imbalance, and is a different notion.
>
> ## Response to Weakness 3
> In our main note to all authors we have updated our work with an algorithm that uses two autoregressive components and requires no gap in the spectrum of the matrix. However, we have added more intuition on understanding the gap in the eigenvalues (see the blue text after Theorem 5).

---

> ### Author Response · Authors · 2024-11-23
> **Responses to Weaknesses (Continued)**
>
> ## Response to Weakness 4
> - Regarding the noise in Figure 5, the high variance is due in particular to bimodality of the accuracies – length generalization is achieved sometimes. We emphasize that resolving this is a modeling/deep learning question that ought to be approached in a task-specific way – our goal was to show that the building block for these STU models (ie the spectral filtering algorithm for LDS) length-generalizes, and to hint that it can be taken advantage of in deep learning tasks. We have added a footnote in blue in the induction heads experimental section to elaborate on the nature of the high variance.
>
> - Building on the above point: as a rough first approximation, one expects transformers to excel in precise copy/lookup tasks like induction heads due to architectural biases, whereas SSMs ought to have a more natural representation of long-range dependency and reasoning-type tasks. It is often difficult to make useful modeling decisions based on synthetic tasks, and it is even more difficult to determine which synthetic task is representative of what we actually care about in application (such as language modeling). As an illustrative example, Mamba (and related methods that introduce a pointwise selection mechanism) was created with the hope of getting the best of both of these memorization and dynamical worlds, and they used length generalization on induction heads to demonstrate this (see Table 2 in [1], where they compare with transformers and other SSMs) – however, we have recently seen in the literature Mamba’s length generalization performance is much more nuanced and exhibits different phenomena on seemingly-similar synthetic tasks (compare Figures 5 and 6 in [2], for example) and in a way that gets even more complicated in language [3]. What can be confidently said is that STU models are capable in many domains (including on these synthetic tasks, though length-generalization was not previously experimented with in STU models) [4], but we feel that it could be misleading or unuseful to focus particularly on comparison with e.g. transformers on synthetic tasks. Our goal was to prove that spectral filtering has robust length generalization, even for marginally-stable systems (the first result of this sort for algorithms that one would use as an SSM layer) and to inspire a proper empirical investigation on large-scale language modeling. If one wants to compare different models side-by-side on synthetic tasks like induction heads, citations [1] and [2] do this for transformers with various positional embeddings (the choice of which also plays a huge role) and SSMs at roughly the same scale as our experiment. However, we believe that these synthetic settings are not quite the right place to make the comparison and designate any SOTA performances on length generalization, and we leave the proper comparison (in language models on long-context evals) to future experimental works.
>
> - [1] – Mamba: Linear-Time Sequence Modeling with Selective State Spaces, Gu & Dao, 2023.
> - [2] – Repeat After Me: Transformers are Better than State Space Models at Copying Transformers are Better than State Space Models at Copying, Jelassi et al., 2024.
> - [3] – DeciMamba: Exploring the Length Extrapolation Potential of Mamba, Ben-Kish et al., 2024.
> - [4] – Flash STU: Fast Spectral Transform Units, Liu et al., 2024

---

> ### Author Response · Authors · 2024-11-23
> **Responses to Questions**
>
> ## Responses to Questions
> 1. To provide some intuition on the gap in the spectrum (which we have now included right after the statement of Theorem 5), here is the following: first imagine that all the eigenvalues of A are bounded by 1-\delta. Then the “weight” that input u_{t-t_0} has for y_t is roughly (1-\delta)^t_0 (because it is multiplied by A t_0 many times). When is this weight small? When t_0 is much larger than 1/\delta. When t_0 is much larger than 1/\delta then u_{t-t_0} doesn’t really impact y_t and so it may be forgotten. This is why we have that if the spectrum lies in [0, 1-log(T)/(8T^q)], length generalization is possible. Indeed, letting \delta = log(T)/8T^q and t_0 = T^q (which is much bigger than 8T^q/log(T) for large enough T) we see that when the spectrum of A is smaller than 1 - \delta, after t_0 = T^q many steps we can forget about the previous inputs u_{t-t0}. For the second part of the range: i.e. that the spectrum of A can lie between [1-1/(2T^{5/4}),1] we note that this is a special feature we prove about spectral filtering (this is rather technical but it is very important because it implies that systems that require long memory are able to be learned with spectral filters). The region between the first part and the second part is exactly that the range where the eigenvalues aren’t small enough that they don’t impact y_t and can be forgotten, but also aren’t large enough that the autoregressive component is able to capture them.
>
> 2. This is a good question, and the empirical evidence sure seems to suggest so. We currently don’t know how to show such theoretical lower bounds.
>
> 3. We plotted these error bars as mean plus/minus 1.96 * standard deviation. The data that generated this plot is very bimodal (either the task is solved or not), and in such a setting it is possible for the error bars to extend past the maximum observation. We figured this indicated the high variance of the data well, and we will add some brief information to the experimental section clarifying the bimodality of the data. If you have any suggestions for a better way to plot this information we can try that out as well, but to us this was the simplest and least cluttered plotting method. As a concrete example, consider a random variable that takes value 1 with probability 0.7 and value 0.5 with probability 0.3. Then, the mean is 0.85 and the standard deviation is 0.229 – the error bars we draw for such a thing would be huge, shading in the interval [0.4, 1.3].

---

### Official Review · Reviewer_yEqw · 2024-11-04

**Soundness:** 2
**Presentation:** 3
**Contribution:** 2
**Rating:** 6
**Confidence:** 3

**Summary:**

The paper considers the problem of sequence prediction of linear dynamical systems, and studies
whether it is necessary to remember the entire history of inputs
to predict optimally. To this end, the paper introduces the notion of asymmetric regret, which
is the regret of the learner compared with a comparator which uses a longer context than the
learner. It is shown that the class of spectral filtering algorithms can guarantee $\sqrt{T}$
asymmetric regret, under suitable assumptions, demonstrating that these existing algorithms can
achieve non-trivial context-length generalization with no modification

**Strengths:**

- The study of context length generalization is interesting and has a lot
of practical relevance, especially given the recent surge of interest in
transformer architectures.
- The paper is generally well-written and easy to follow

**Weaknesses:**

- The asymmetric regret seems to be a *weaker* notion than usual regret; often regret guarantees
hold for an *arbitrary* comparator, which would include comparators that utilize more information
than the learner as a special case. Yet
it is claimed on lines 258/259 that "The comparator can observe more information. Thus, this is a stronger, more difficult guarantee to obtain than classical regret."

- The experimental results in figure 5 seem flawed; the shaded regions
imply that the models can reach accuracy above 100%. I assume the
shaded region must be $\pm$ some measure of spread (unspecified),
but this doesn't make sense when it extends outside the domain of the
measurement.

**Questions:**

- Why do we necessarily care about competing with the best LDS predictor?
One receives the input/features before predicting, making it possible to
make "improper" predictions. For instance, the Vovk-Azoury-Warmuth forecaster
guarantees logarithmic static regret without any assumptions on the data
by leveraging improper predictions. Analogously, it could be the case that
the best LDS predictor is much worse than the best improper predictor.

- I don't understand how the problematic eigenvalues of A are non-extremal ones; what
exactly is special about the upper and lower bounds, that causes problems
in the range between them?

---

> ### Author Response · Authors · 2024-11-23
> **Response to Weaknesses**
>
> ## Responses to Weaknesses
> 1. We first note that we have now added an extra paragraph (highlighted in blue) on asymmetric-regret (right after definition 3) to help ensure extra clarity based on this point. Please let us know if you find it to be helpful. Asymmetric regret is a stronger notion because it implies regular regret. The confusion might be with comparator classes and when those are fixed. In sequence prediction literature, regular regret considers comparator classes in which both the learner and the comparators construct their predictions based on the same sequence length. With asymmetric regret, we measure the learner against even stronger comparators which are allowed to construct their predictions using longer sequence lengths (and therefore more information). Your statement that “often regret guarantees hold for an arbitrary comparator” isn’t accurate. Indeed, if you are comparing an algorithm’s performance to an arbitrary comparator then you will often get meaningless results since you can always pick a predictor which predicts perfectly. For this reason, you will find that regret results in the sequence prediction literature have been based on a restricted comparator class. To understand the difference between asymmetric and regular regret better, fix any comparator class in sequence prediction and you can consider regular regret where the comparator class must use the same information as the learner, or you can consider asymmetric regret where the comparator class is allowed to use more information. In other words, competing against an adversary that has more information than the learner is harder than competing against an adversary that has access to the exact same information that the learner has. Therefore, a guarantee against the stronger adversary automatically implies a guarantee against the weaker one. This is a central concept in our paper, so please ask again if we can clarify further.
>
> 2. We plotted these error bars as mean plus/minus 1.96 * standard deviation. The data that generated this plot is very bimodal (either the task is solved or not), and in such a setting it is possible for the error bars to extend past the maximum observation. We figured this indicated the high variance of the data well, and we will add some brief information to the experimental section clarifying the formula for the error bars as well as the bimodality of the data. If you have any suggestions for a better way to plot this information we can try that out as well, but to us this was the simplest and least cluttered plotting method.
> As a concrete example, consider a random variable that takes value 1 with probability 0.7 and value 0.5 with probability 0.3. Then, the mean is 0.85 and the standard deviation is 0.229 – the error bars we draw for such a thing would be huge, shading in the interval [0.4, 1.3].

---

> > ### Comment · Reviewer_yEqw · 2024-11-24
> >
> > 1. I see, so you are saying that asymmetric regret is a stronger notion than *static* regret. But asymmetric is still a strictly weaker notion that dynamic regret, which *does* allow comparison to an arbitrary comparator sequence. What is being afforded here by asymmetric regret that couldn't already be expressed using dynamic regret as a special case? Any algorithm making dynamic regret guarantees in this setting would also make asymmetric regret guarantees as well, would it not?
> >
> > >Your statement that “often regret guarantees hold for an arbitrary comparator” isn’t accurate. Indeed, if you are comparing an algorithm’s performance to an arbitrary comparator then you will often get meaningless results since you can always pick a predictor which predicts perfectly. For this reason, you will find that regret results in the sequence prediction literature have been based on a restricted comparator class.
> >
> > This response seems to confuse competing with an arbitrary comparator with competing with a specific arbitrary one. Regret guarantees often hold for *all comparators simultaneously*, $R_T(u)\le F(u)$ for all $u$. Typically there is a term which expresses the complexity of the comparator, such as $\\|u\\|$ for static regret or the path-length $\sum_t\\|u_t-u_{t-1}\\|$ for dynamic regret. It is true that the bound usually becomes vacuous when you plug in a perfect predictor, but this is usually not the one you would actually care about (e.g. you would rather compare against a comparator that accurately predicts the mean of the distribtion generating $y_t$)
> >
> > 2. I don't think this explanation fixes my concern; yes it is possible for the error bars to exceed the maximum value if you plot $\pm$ standard deviation, but this means you are not plotting a meaningful measure of spread. The plot should be indicating uncertainty in the random variable being measured, and you know *for sure* that the variable will not be greater than a certain value, so plotting $\pm$ the standard deviation is not meaningfully expressing this uncertainty

---

> > > ### Author Response · Authors · 2024-11-24
> > > **Regret notions in online learning**
> > >
> > > 1. The notion of dynamic regret, similar to static regret, doesn't capture information asymmetry, which is the main topic of our paper.  Dynamic regret compares decisions from the same decision set, whereas we compare different predictor classes.
> > > 2. It is standard in scientific graphs to plot confidence intervals, even though it's not tight and bounds above 1 or below 0 are not meaningful. If you have a different suggestion we are happy to do it!

---

> > > > ### Comment · Reviewer_yEqw · 2024-11-24
> > > >
> > > > 1. I am not sure I follow. Both dynamic regret and asymmetric regret require the decisions to come from a common decision space. It cannot be the case that the learner chooses $x_t\in\mathcal{X}\subset\mathbb{R}^d$ while the comparator is allowed to choose iterates outside of the decision set $\mathcal{X}$ --- in such settings the learner can always be forced to incur linear regret.
> > > > Where dynamic regret and asymmetric regret differ is that asymmetric regret further constrains the comparator sequence to some specific subset of sequences from $\mathcal{X}$ (namely, sequences in $\mathcal{X}$ that use more history than the learner). But this is ultimately just a special case of dynamic regret, since in dynamic regret we are free to consider *any* sequence in $\mathcal{X}$, including those which use more information than the learner. Moreover, since dynamic regret guarantees hold for arbitrary comparator sequences, any algorithm minimizing dynamic in this setting will necessarily also minimize asymmetric regret.
> > > > 2. It was stated in the original reply that the data was bimodal, in which case reporting confidence intervals becomes quite significantly more complicated so I'm not sure what to suggest, as the decision will ultimately require more specifics about the distribution. Perhaps you could consider bootstrap confidence intervals, or simply gathering more sample estimates by conducting a larger number of indepndent runs of your experiment. But in any case, the fact that the data is bimodal suggests that this boundary issue *is* meaningfully effecting the distribution of outcomes in this case, in such a way that you can not just plot mean $\pm$ standard deviation as a valid measure of spread.

---

> > > > > ### Author Response · Authors · 2024-11-24
> > > > > **reply to further discussion**
> > > > >
> > > > > thanks for the engagement!
> > > > >
> > > > > 1. What you wrote is not correct in certain cases:
> > > > > a) when the decision set is that of policies: decisions do not have to come from the same world, and yet there is no linear lower bound on the regret. In our case, we have policies that use different information (context lengths), and yet we can compete with a stronger policy class than the one that the learner optimizes.
> > > > > b) when there is an assumption about the generation of the underlying sequence and it's not adversarial, such as in our case.
> > > > >
> > > > > It may be possible to frame asymmetric regret as a special case of dynamic regret (it's not straightforward), but we think this is not
> > > > > helpful to the general reader, except maybe a specialized expert,  since both the motivation and application are different.
> > > > >
> > > > > 2. we can add a comment that bounds above 1 are meaningless, as you point out, and we can also clip above 1 if that's preferable.

---

> > > > > > ### Comment · Reviewer_yEqw · 2024-11-24
> > > > > >
> > > > > > I think the difference between the decision set and the comparator class is still being confused here. Even in the case of policies, there is a shared decision set: mappings from histories to actions. Dynamic regret over policies compares the loss incurred by the learner's sequence of policies to that of any other sequence of policies. The difference being now that the comparator policy is allowed to change in a data-dependent way. I still do not see how the asymmetric regret is providing insights that are not already provided by the existing notion.

---

> > > > > > > ### Author Response · Authors · 2024-11-25
> > > > > > > **further reply**
> > > > > > >
> > > > > > > We respectfully think that existing notions do not deal with information asymmetry, and rather with changing environments.  Perhaps we are deviating from the main topic of provable length generalization,  is there a specific question we can answer?

---

> > > > > > > > ### Comment · Reviewer_yEqw · 2024-11-25
> > > > > > > >
> > > > > > > > Yes, my question was "What is being afforded here by asymmetric regret that couldn't already be expressed using dynamic regret?", The answer appears to be "nothing, but for the purposes of this paper it is useful to assign a specific semantic meaning to the subset of comparator sequences we'd like to compare against"

---

> > > > > > > > > ### Author Response · Authors · 2024-11-25
> > > > > > > > > **we agree**
> > > > > > > > >
> > > > > > > > > this is a reasonable summary :-)

---

> > > > > > > > > > ### Comment · Reviewer_yEqw · 2024-11-26
> > > > > > > > > >
> > > > > > > > > > So then is there a compelling reason that the results can't be obtained as a special case of existing dynamic regret guarantees?

---

> > > > > > > > > > > ### Author Response · Authors · 2024-11-26
> > > > > > > > > > > **important clarification**
> > > > > > > > > > >
> > > > > > > > > > > Absolutely. Our result are by no means "obtained as a special case" of any definition of regret.  The main result, proving length generalization for spectral filtering, involves several nontrivial technical steps as outlined in other responses, we believe it is a deep and meaningful theorem, the first of its kind anywhere. The variant of regret used is the *outcome* of our investigation, and not a consequence of a choice of any particular metric of performance. The definition of asymmetric regret is not a main contribution, it is a tool to highlight the result.  Before we continue discussing regret variants, are we in agreement on this point?

---

> > > > > > > > > > > > ### Comment · Reviewer_yEqw · 2024-11-26
> > > > > > > > > > > >
> > > > > > > > > > > > I think the discussion of regret variants is concluded; we seem to now agree that asymmetric regret guarantees are less general than dynamic regret guarantees, which we agree is fine for the purposes of this paper.
> > > > > > > > > > > >
> > > > > > > > > > > > Then, the follow-up question was whether the results here can be inferred from the more general guarantees, which hold for *any* comparator sequence, and whether there is a compelling reason that this could not be the case.
> > > > > > > > > > > >
> > > > > > > > > > > > To make things more concrete, Baby & Wang (2022) for instance provide algorithms which guarantee dynamic regret $R_T\le O\left(T^{1/3}C_T^{2/3}\right)$, where $C_T=\sum \|\mu_t-\mu_{t-1}\|$ is the variation of an *arbitrary* comparator sequence of predictions.  Is it necessarily the case that $C_T\le O(T^{1/4})$ is never true under your assumptions? If not, then their  $T^{1/3}C_T^{2/3}$ bound produces the $\tilde O(\sqrt{T})$ bound presented here as a special case.
> > > > > > > > > > > >
> > > > > > > > > > > > - Baby, Dheeraj, and Yu-Xiang Wang. "Optimal dynamic regret in LQR control." Advances in Neural Information Processing Systems 35 (2022): 24879-24892.

---

> ### Author Response · Authors · 2024-11-23
> **Responses to Questions**
>
> ## Responses to Questions
> 1. First, notice that our method is indeed improper - it doesn’t do system identification to create an LDS predictor, but rather predicts using spectral filters. We must have a benchmark class in regret, since otherwise the problem is too hard, this is a consequence of Kolmogorov’s theorem. To answer your question more clearly: 1) When the sequence is a linear dynamical system, then the best improper predictor can’t outperform the best  LDS predictor since the sequence is in fact an LDS and therefore the best LDS predictor can predict it exactly. 2) However, you are right that improper learning algorithms are a good tool and indeed we use an improper learning algorithm. 3) Perhaps the reviewer has some confusion regarding the guarantee afforded by the Vovk-Azoury-Warmuth (VAW) forecaster( see for example Theorem 2 in this blogpost https://parameterfree.com/2019/10/17/follow-the-regularized-leader-iii-more-logarithmic-bounds/). Please note that the VAW forecaster makes improper predictions by itself but it affords a regret bound only against the class of fixed (but potentially unbounded) linear predictors, i.e. the VAW forecaster only provides a guarantee against any predictor of the type that predicts as the rule hat{y}_t =  <u_t, x> where x is held fixed at every time step. In comparison, the class of LDS predictors is significantly larger as every prediction can take each of the preceding tokens into account as opposed to only the current token. To see this consider an LDS with no state, i.e. A = 0: this extremely special case of an LDS captures every fixed linear predictor. Thus while the VAW forecaster provides a logarithmic regret bound on a potentially unbounded class of predictors, these predictors are restricted to be history-less linear predictors which is a much more restricted class of predictors than LDS predictors against which our algorithm provides a regret guarantee.
>
> 2. Inspired by this question, we have now added a discussion of the problematic eigenvalues (highlighted in blue) directly after Theorem 5. We note that we now have an algorithm that is robust to these problematic eigenvalues. However, to provide some intuition on the gap in the spectrum (which we have now included right after the statement of Theorem 5), here is the following: first imagine that all the eigenvalues of A are bounded by 1-\delta. Then the “weight” that input u_{t-t_0} has for y_t is roughly (1-\delta)^t_0 (because it is multiplied by A t_0 many times). When is this weight small? When t_0 is much larger than 1/\delta. When t_0 is much larger than 1/\delta then u_{t-t_0} doesn’t really impact y_t and so it may be forgotten. This is why we have that if the spectrum lies in [0, 1-log(T)/(8T^q)], length generalization is possible. Indeed, letting \delta = log(T)/8T^q and t_0 = T^q (which is much bigger than 8T^q/log(T) for large enough T) we see that when the spectrum of A is smaller than 1 - \delta, after t_0 = T^q many steps we can forget about the previous inputs u_{t-t0}. For the second part of the range: i.e. that the spectrum of A can lie between [1-1/(2T^{5/4}),1] we note that this is a special feature we prove about spectral filtering (this is rather technical but it is very important because it implies that systems that require long memory are able to be learned with spectral filters). The region between the first part and the second part is exactly that the range where the eigenvalues aren’t small enough that they don’t impact y_t and can be forgotten, but also aren’t large enough that the autoregressive component is able to capture them. It’s crucial that the problematic eigenvalues are the non-extremal ones, and that they can be solved with a  constant amount of extra computation (ie the second autoregressive term). In most algorithms, the problematic eigenvalues would be the marginally-stable ones (i.e. the ones that approach 1), and there is nothing that can be done about those. This is why the length generalization of spectral filtering is nontrivial at all.

---

> > ### Comment · Reviewer_yEqw · 2024-11-24
> >
> > >Perhaps the reviewer has some confusion regarding the guarantee afforded by the Vovk-Azoury-Warmuth (VAW) forecaster
> >
> > This was just given as an example of how improper predictors can often have additional nice properties, such as competing against an arbitrary comparator and no assumptions on the data. Perhaps for this paper, a more directly relevant example would have been the extension of the VAW forecaster from Jacobsen & Cutkosky (2024), which makes guarantees under the same conditions as the VAW forecaster but for dynamic regret, so that the guarantees hold against an arbitrary comparator sequence. So my question still stands: why do we necessarily care about competing with the best LDS comparator?
> >
> > ## References
> >
> > - Jacobsen, A., & Cutkosky, A. (2024). Online linear regression in dynamic
> >   environments via discounting. In , Proceedings of the 41st International
> >   Conference on Machine Learning (pp. 21083–21120). : PMLR.

---

> > > ### Author Response · Authors · 2024-11-24
> > > **why LDS are interesting**
> > >
> > > The answer is that LDS are natural comparator classes, they are very expressive with a high hidden dimension, they admit theoretical analysis, and they are standard in the literature. In contrast, even simple nonlinear dynamical systems are intractable to analyze in general (see ref below). For this reason extensive previous literature in sequence prediction (Griffin, LRU, Hippo, spectral filtering) relate to LDS, and so do we. More justification for the merit of LDS can be found here:
> > > - Hazan, Singh, Introduction to Online Nonstochastic Control

---

> ### Author Response · Authors · 2024-11-26
> **Understanding regret variants and the distinction between dynamic and asymmetric regret**
>
> We apologize for our message stating "this is a reasonable summary", this was wrong and we do not agree on regret variants. In our paper, we ask the question of whether there are algorithms which form predictions based on a class of functions which use the previous $L$ inputs and compare nicely had they used the previous $T$ inputs (for $L << T$).
> Dynamic regret looks at the best arbitrary sequence of comparitors in hindsight. For $x_t \in \mathcal{D}$ we consider:
> $$\textrm{Dynamic Regret} = \sum_{t = 1}^T f_t(x_t) - \min_{(w_1, \dots, w_t) \in \mathcal{D}} \sum_{t = 1}^T f_t(w_t).$$
> Let’s see why dynamic regret cannot capture asymmetric regret. Let’s suppose the input (note that this is a fixed input vector that comes from the data) $u_t$ is the $t^{\textrm{th}}$ basis vector $e_t$ (all zeros vector of length $T$ with $1$ at the $t^{\textrm{th}}$ element) and suppose we consider linear predictors of the form (constrained by context length $L$):
> $$\sum_{i = t-L}^T V_{i}^{\top} e_{i}.$$
> Therefore the domain is $x_t = (V_{t-L,t}, ..., V_{L,t}) $ where each $V_{i, t}$ a diagonal $TxT$ matrix (note if $t-L <0$ then we set it to $0$).
> What is the standard (static) regret in this case? First you fix what $L$ is and you consider:
> $$\textrm{Regret} = \sum_{t = 1}^T f_t\left( x_t \right) - \min_{ w}  \sum_{t = 1}^T f_t(w).$$ We assume that
> $$f_t(x_t) = \ell_t\left( \sum_{i = t-L}^T V_{i}^{\top} e_{i} \right).$$ Thus we have
> $$\textrm{Regret} =  \sum_{t = 1}^T \ell_t \left( \sum_{l = t-L}^t V_{i,t}^{\top} e_{i}\right) - \min_{ (B_{1}, ..., B_T)} \sum_{t = 1}^T \ell_t(\sum_{l = t-L}^t B_{i}^{\top} e_{i}).$$
> What is dynamic regret? Now the choice of matrices B can depend on t:
> $$ \textrm{Dynamic Regret} = \sum_{t = 1}^T \ell_t \left( \sum_{l = t-L}^t V_{i,t}^{\top} e_{i} \right) - \min_{ (B_{1,1}, ..., B_{T,1}), \dots, (B_{1,T}, \dots, B_{T,T})} \sum_{t = 1}^T \ell_t \left( \sum_{l = t-L}^t B_{i,t}^{\top} e_{i} \right).$$
> What is asymmetric regret? We now want to compare to predictors which are allowed to use length T>>L , so the sum in the comparison starts at 1 instead of t-L:
> $$ \textrm{Asymmetric Regret} = \sum_{t = 1}^T \ell_t \left( \sum_{l = t-L}^t V_{i,t}^{\top} e_{i} \right) - \min_{ (B_{1}, ..., B_T)} \sum_{t = 1}^T \ell_t \left( \sum_{l = 1}^t B_{i}^{\top} e_{i}\right).
> $$
> (Notice that this CANNOT be captured by dynamic regret since $ \sum_{t = 1}^T \ell_t \left( \sum_{l = t-L}^t V_{i,t}^{\top} e_{i} \right)$ is supported only on the last $L-T$ coordinates (the matrices are diagonal and the inputs are the basis vectors) whereas $sum_{l = 1}^t B_{i}^{\top} e_{i}$ can be supported on all the coordinates.
> Notice that we could also consider asymmetric and dynamic regret:
> $$ \textrm{Asymmetric and Dynamic Regret} = \sum_{t = 1}^T \ell_t \left(  \sum_{l = t-L}^t V_{i,t}^{\top} e_{i} \right) - \min_{ (B_{1,1}, ..., B_{T,1}), \dots, (B_{1,T}, \dots, B_{T,T})} \sum_{t = 1}^T \ell_t \left( \sum_{l = 1}^t B_{i,t}^{\top} e_{i} \right).$$
>
>  The fact that asymmetric regret allows a different domain for the predictor and the comparator class means there are instances (depending on the comparator class) where linear regret can always be incurred (as you noted!). However, when you constrain to the same class of predictors then (as we show in our paper) there can surprisingly be hope that you get $O(\sqrt(T))$ regret. Indeed, in our paper we consider predictors of the form:
>  $y_t = \sum_{i = T-L}^T M_{i} u_{i}$ and the comparator class is $y_t = \sum_{i = 1}^T B_{i} u_{i}$, where $B_{i}$ has a special form based on spectral filtering that is shown to be (1) computable and (2) have optimal performance compared to the best LDS predictor in hindsight.
>
> Finally, we hope this response made everything clear but if you are still skeptical we recommend looking at the algorithm in the Baby & Wang (2022) paper and formally spelling out what the domain is and what the algorithm would be outputting and how this would NOT capture our notion of asymmetric regret and therefore their results DO NOT apply to our work (indeed either the predictors would be allowed to base their predictions on full context length or the comparator class would have shortened context length using their definitions and depending on how you want to define the domain, neither of which are acceptable for the purposes of our paper). We appreciate this reviewer's carefulness in understanding the distinctions between these things and we recognize the subtlety.

---

> > ### Comment · Reviewer_yEqw · 2024-11-26
> >
> > The response seems to be confusing the definition of dynamic regret with the special case where you consider the worst-case comparator sequence. Dynamic regret need not be defined as being against the sequence of comparators that minimize the cumulative loss, but can more generally be defined as the regret against *any* sequence of comparators:
> > $$
> > R_T(u_1,\ldots,u_T) = \sum_t \ell_t(w_t)-\ell_t(u_t).
> > $$
> > There are algorithms, including those in the papers cited earlier, which make guarantees against *any* sequence we'd like. This means we can, as a special case, set them to *exactly the same sequence as the one that was defined above* which you are claiming distinguishes asymmetric regret from dynamic regret.

---

> > > ### Author Response · Authors · 2024-11-26
> > >
> > > The reviewer is simply stating that you can consider the regret over each comparator in the class rather than taking the min. Our statements still hold and we urge the reviewer again to carefully go through the paper they cite to see how these concepts are different.

---

> > > > ### Comment · Reviewer_yEqw · 2024-11-26
> > > >
> > > > No, I am stating that you can consider the dynamic regret *against an arbitrary sequence of comparators*, and if you can make a guarantee for *all sequences simultaneously*, as existing algorithms do, you necessarily must also have a guarantee against comparator sequences which use less information than the learner as a special case. In your example, I am freely able to compare the learner's cumulative loss against that of the comparator sequence $u_t=\sum_{\ell=1}^t B_\ell^\top e_\ell$.

---

> > > > > ### Author Response · Authors · 2024-11-26
> > > > >
> > > > > This is incorrect. Again, dynamic regret assumes the class of comparators and the class the algorithm can return from is the same. Indeed you can consider an arbitrary sequence of comparators, however they must be from the same class. Please look at our example again. If we compare against $u_t = \sum_{\ell = 1}^t B_{\ell}^{\top} e_{\ell|}$, then we are allowed to output things that have support over all of $R^T$, meaning the algorithm also must be allowed to output predictions of the form $x_t = \sum_{\ell = 1}^t B_{\ell}^{\top} e_{\ell}$ and thus is not context-length constrained. Please read our example again and make sure you understand that dynamic regret just means that the choice of matrices $B_{\ell}$ are now allowed to depend on $t$:
> > > > > $B_{\ell,t}$.

---

> > > > > > ### Comment · Reviewer_yEqw · 2024-11-26
> > > > > >
> > > > > > You are incorrect, the constraints you are referencing are self imposed by the algorithm itself, not the problem setting of sequence prediction. The sequences compared against must be from the same base *decision set* in dynamic regret, e.g., if $x_t\in \mathbb{R}^d$ for all t then we must also have $u_t\in\mathbb{R}^d$ for all t. There is no such restriction that they must come from the same subset of sequences $\mathbb{R}^d$ if the learner chooses to restrict themselves to context-constrained sequences, so we are indeed free to choose $u_t$ in the same way you would for asymmetric regret and it would still be a valid expression of dynamic regret.

---

> > > > > > > ### Comment · Reviewer_yEqw · 2024-11-26
> > > > > > >
> > > > > > > Perhaps the issue is that we're talking about two separate but related issues. First, I assume we can agree that the set of all comparator sequences in $\mathbb{R}^d$ contains the set of all comparator sequences in $\mathbb{R}^d$ restricted to a certain memory constraint condition, yes? This means dynamic regret is more general than asymmetric regret. Take any sequence of losses $\ell_1,\ldots,\ell_T$, dynamic regret just compares the learner's total loss against that of any other sequence of inputs to the losses. In the context of sequence prediction, this means we can compare against any possible sequence of predictions $\hat y_t$ and it would be valid dynamic regret expression, including any sequence you claim makes asymmetric regret distinct. Surely there is nothing controversial about this statement.
> > > > > > >
> > > > > > > The second issue was the question "can this result be produced as a special case of *existing dynamic regret guarantees*?". This seems to be what you are answering with the example: the existing dynamic regret *guarantees* hold for an arbitrary comparator sequence, but there's no guarantee that the algorithm that achieves those guarantees will choose memory-constrained iterates, and hence the guarantees here can not necessarily be produced as a special case of those existing algorithms. Do we agree on this interpretation?

---

> > > > > > > > ### Author Response · Authors · 2024-11-26
> > > > > > > >
> > > > > > > > This is completely wrong. We reiterate to the reviewer to please read through our example carefully. We add a few things that might be helpful from our detailed answer above. 1) The domain is NOT R^d (note this isn't even constrained), the domain is the set of matrices the algorithm gets to choose and would typically have a constraint on its norm (we think the reviewer believes the domain is just R^d, i.e. the output vector). This is standard in online learning literature, i.e. that the algorithm outputs a vector which is a function of its domain and maybe the overloaded notation in the literature is causing the confusion.  2) In dynamic regret you must assume $x_t$ is in the same domain as $w_1$, ..., $w_T$ (i.e. the set of matrices the algorithm is allowed to pick to construct its output is the same as the set of matrices it compares itself against). We will edit our detailed response with this change and hope that it helps clear things up for the reviewer.
> > > > > > > >
> > > > > > > > We unfortunately do not feel that this has been productive and although we appreciate the reviewer's interest and have good faith they want to understand, we don't know how to further explain these distinctions. We again ask the reviewer to please read what we wrote and to look through the paper they cited to understand that the algorithm in that paper would not produce context length restricted predictions.

---

> > > > > > > > > ### Comment · Reviewer_yEqw · 2024-11-26
> > > > > > > > >
> > > > > > > > > > We again ask the reviewer to please read what we wrote and to look through the paper they cited to understand that the algorithm in that paper would not produce context length restricted predictions
> > > > > > > > >
> > > > > > > > > This was quite precisely the content of the second paragraph of my previous response, please read my responses in full before claiming them to be "completely wrong".
> > > > > > > > >
> > > > > > > > > You are correct that I was being careless with the domain and defaulted to the simple case of $x_t\in\mathbb{R}^d$, my apologies if this was upsetting. The point still stands: there are loss functions, and the learner chooses inputs to the loss functions. The domain of the loss functions is the common set that both the learner and the comparator sequence must both share. There is no meaningful way to introduce information asymmetry other than by imposing additional restrictions on top of this setup, such as by constraining to a subset of possible sequences that the algorithm or comparator sequence will input into the loss functions, or by imposing restrictions on what strategies the learner and comparator are allowed to use to predict $y_t$.
> > > > > > > > >
> > > > > > > > > I will retain my original (positive) score.

---

### Author Response · Authors · 2024-11-23
**Main Note to all Reviewers**

We thank the reviewers for their insightful comments and questions. The reviewers feedback has certainly resulted in a clearer paper which better highlights the strength of the results and exciting potential implications. In the new version of the paper, we highlight the changes we made in direct response to a reviewer comment in blue. We have two very important comments for all reviewers so please make sure to read below.

**Enhanced result:** One of the main weaknesses in the theoretical results established by our paper was the assumption on the spectrum of the system matrix A. Our proof technique (which we discuss next) provided us with an algorithmic insight into how to mitigate sensitivity to the “bad eigenvalue range”. Our paper now includes an entirely novel algorithm based on spectral filtering which surprisingly shows that simply by using one more autoregressive component, it is possible to achieve length generalization that is completely robust to the spectrum of A. Moreover, we show experimentally that, while the original spectral filtering algorithm introduced in previous literature performs poorly on this small bad range of eigenvalues, our new algorithm performs well (as our theory predicts).

**Nontriviality of proof:**  Several of the reviewers were under the assumption that the results of this paper come from synthesizing previous results. This misapprehension is a fault in our writing– the proof of length generalization is not at all trivial and is not simply an application of previous work. Not only is it nontrivial, it is surprising and exciting. Perhaps in our effort to make the result seem approachable, we diminished what it entails. We have edited section 3.1 (see the text highlighted in blue) to make this clearer. Here are the key nontrivial and completely novel elements that are involved in our proof:
1) We prove that minimizing the loss presented in our algorithms results in approximate recovery of the best (and untruncated) spectral filtering predictor – this requires proving that the loss function is strongly convex and proving uniqueness of the best untruncated (i.e. choose k = T rather than O(log(T))) spectral filtering predictor.
2) We prove that the best untruncated spectral filtering predictor achieves length generalization (this is where our insight that using two autoregressive components changes the assumptions needed on the spectrum of A).
3) Finally we provide error bounds to show that even though we set k = log(T) rather than k = T and even though we don’t converge exactly to the best untruncated spectral filtering predictor (this error depends on the strong convexity constant), we are still able to approximately enjoy the length generalization guarantees proven in part 2.

For clarity, NONE of these elements existed in previous papers, they are completely novel, and they required different insights. Moreover, this proof technique inspired our new algorithm which was never presented before, which uses the two autoregressive components to achieve more robust length generalization.

***Thank you all for the constructive responses, we are very grateful and would be more than happy to continue any discussions and incorporate any more feedback. Have a lovely weekend!***

---

> ### Comment · Reviewer_yEqw · 2024-11-24
>
> Since the authors were willing to upload a revised draft, could you post one with the proper ICLR 2025 stylefile? the margins on this paper were already modified to allow significantly more space than the submissions were afforded, and now the current draft even exceeds the page limit on top of that. I think it would be unfair to other submissions to change my score based on revisions that would far exceed the page limit guidelines that all other submissions had to abide by.

---

> > ### Author Response · Authors · 2024-11-24
> >
> > The current draft should no longer exceed the page limit. We aren't sure what you mean by it not having the proper ICLR stylefile, since that is what we are using and we didn't change anything (the margins for instance). If there is something we are missing or if we have somehow used the wrong ICLR 2025 package then of course we are happy to change it!

---

> > > ### Comment · Area_Chair_bv5r · 2024-11-24
> > >
> > > Dear authors,
> > >
> > > Could you please double check if you are using the correct ICLR stylefile?  The margins of the current draft still do not look correct, and it does not include the line "Under review as a conference paper at ICLR 2025".

---

> > > > ### Author Response · Authors · 2024-11-24
> > > >
> > > > We had accidentally loaded the \usepackage{fullpage} and fixed it!

---

> > > > > ### Comment · Reviewer_yEqw · 2024-11-24
> > > > >
> > > > > Could you briefly describe what was changed in the paper to make it fit within the page limit?

---

> > > > > > ### Author Response · Authors · 2024-11-25
> > > > > >
> > > > > > We changed some sentences so they were a bit more terse, we made figure captions more brief, and we made the induction heads plot a bit smaller. There wasn't too much that needed to be changed.

---

### Meta-Review · Area_Chair_bv5r · 2024-12-24

**Metareview:**

This paper studies length generalization in sequence prediction and defines a new metric of performance called asymmetric regret, which measures the regret compared with a predictor with longer context length than available to the learner. Under this new metric, the authors show that an algorithm based on spectral filtering attains good asymmetric regret under certain technical assumptions. Experiments results are presented to validate the theoretical findings.

This is a borderline submission. The main weakness raised by the reviewers is the assumption on the spectrum of the system matrix. Although the authors later provided a new algorithm that might get rid of such assumption, another review cycle might be necessary to fully validate these new results. Another weakness raised by the reviewers is the technical novelty. Although the authors provided a summary of the novel parts of their proof, it is still unclear if the reviewers' concern has been addressed.

Given all the factors mentioned above and the high standards of ICLR, I lean towards rejecting this submission.

**Additional Comments On Reviewer Discussion:**

The reviewers raised concerns regarding the technical novelty of the paper, the assumption on the spectrum of the system matrix, as well as comparison between asymmetric regret and dynamic regret,. Although the authors provided responses which addressed some of those concerns, another review cycle might be necessary to fully validate the new algorithm that is robust to the spectrum of the system matrix, and it is still unclear if the reviewers' concern regarding the technical novelty of the paper has been addressed.

---

### Decision · Program_Chairs · 2025-01-22

Reject